# Longitudinally monitored immune biomarkers predict the timing of COVID-19 outcomes

Gorka Lasso[1], Saad Khan[2], Stephanie A. Allen[3,¤a], Margarette Mariano[4], Catalina Florez[1,5,¤b,¤c], Erika P. Orner[6], Jose A. Quiroz[3], Gregory Quevedo[4], Aldo Massimi[4], Aditi Hegde[7], Ariel S. Wirchnianski[1,4], Robert H. Bortz, III[1], Ryan J. Malonis[4], George I. Georgiev[4], Karen Tong[4], Natalia G. Herrera[4], Nicholas C. Morano[4,¤d], Scott J. Garforth[4], Avinash Malaviya[3], Ahmed Khokhar[3], Ethan Laudermilch[1,¤e], M. Eugenia Dieterle[1], J. Maximilian Fels[8,9,10], Denise Haslwanter[1], Rohit K. Jangra[1,¤f], Jason Barnhill[5,11], Steven C. Almo[4], Kartik Chandran[1]*, Jonathan R. Lai[4]*, Libusha Kelly[1,2]*, Johanna P. Daily[1,3]*, Olivia Vergnolle[4,¤g]*

1 Department of Microbiology and Immunology, Albert Einstein College of Medicine, Bronx, New York, United States of America, 2 Department of Systems and Computational Biology, Albert Einstein College of Medicine, Bronx, New York, United States of America, 3 Division of Infectious Diseases, Department of Medicine, Albert Einstein College of Medicine and Montefiore Medical Center, Bronx, New York, United States of America, 4 Department of Biochemistry, Albert Einstein College of Medicine, Bronx, New York, United States of America, 5 Department of Chemistry and Life Science, United States Military Academy at West Point, West Point, New York, United States of America, 6 Department of Pathology, Albert Einstein College of Medicine, Bronx, New York, United States of America, 7 Eastchester High School, 2 Stewart Place, Eastchester, New York, United States of America, 8 Department of Cell Biology, Harvard Medical School, Boston, Cambridge, Massachusetts, United States of America, 9 Department of Microbiology, Harvard Medical School, Boston, Cambridge, Massachusetts, United States of America, 10 Department of Cancer Immunology and Virology, Dana-Farber Cancer Institute, Boston, Cambridge, Massachusetts, United States of America, 11 Department of Radiology and Radiological Services, Uniformed Services University of the Health Sciences, Bethesda, Maryland, United States of America

☯ These authors contributed equally to this work.
¤a Current address: Department of Internal Medicine, Yale New Haven Hospital and Yale New Haven Health System, New Haven, Connecticut, United States of America
¤b Current address: U.S. Army Medical Research Institute of Infectious Diseases, Frederick, Maryland, United States of America
¤c Current address: The Geneva Foundation, Tacoma, Washington, United States of America
¤d Current address: Zuckerman Institute, Columbia University, New York, New York, United States of America
¤e Current address: 3M Corporate Research Materials Laboratory, Saint Paul, Minnesota, United States of America
¤f Current address: Department of Microbiology and Immunology, Louisiana State University Health Science Center-Shreveport, Shreveport, Louisiana, United States of America
¤g Current address: Tri-Institutional Therapeutics Discovery Institute, New York, New York, United States of America
* kartik.chandran@einsteinmed.org (KC); jon.lai@einsteinmed.org (JRL); libusha.kelly@einsteinmed.org (LK); jdaily@montefiore.org (JPD); olivia.vergnolle@einsteinmed.org (OV)

**Data Availability Statement:** All the codes used for this study can be accessed on GitHub at the

## Abstract

The clinical outcome of SARS-CoV-2 infection varies widely between individuals. Machine learning models can support decision making in healthcare by assessing fatality risk in patients that do not yet show severe signs of COVID-19. Most predictive models rely on static demographic features and clinical values obtained upon hospitalization. However,

following address: https://github.com/kellylab/longitudinal-prediction-of-covid-outcomes.

**Funding:** This work was supported by the National Institutes of Health R01AI132633 to K.C., R01-AI125462 to J. R. L. and R21AI141367 to J.P.D (https://www.nih.gov). L.K. is supported in part by a Peer Reviewed Cancer Research Career Development Award from the United States Department of Defense (CA171019) (https://cdmrp.army.mil/funding/prcrp). C.F. was supported by an NRC Research Associateship award (https://sites.nationalacademies.org/PGA/RAP/index.htm). M.E.D. is a Latin American Fellow in the Biomedical Sciences, supported by the Pew Charitable Trusts (https://www.pewtrusts.org/en/projects/biomedical-research). S.K., R.H.B.III., and R.J.M. were partially supported by the National Institutes of Health training grant 2T32GM007288-45 (Medical Scientist Training Program) at Albert Einstein College of Medicine (https://www.nigms.nih.gov/training/instpredoc/pages/predocoverview-mstp.aspx). Saad Khan (S.K) was supported an NIH T32 fellowship on Geographic Medicine and Emerging Infectious Diseases (2T32AI070117-13) (https://www.niaid.nih.gov/about/infectious-diseases-fellowship-application-information), and a grant from the Ullmann Family Foundation. Libusha Kelly (L.K) wassupported in part by an award from the Google Cloud Research Credits program (GCP19980904) (https://cloud.google.com/edu/researchers). J.R.L and K.C. were recipients of a COVID-19 pilot project grant from the Albert Einstein College of Medicine. The funders had no role in study design, data collection and analysis, decision to publish, or preparation of the manuscript.

**Competing interests:** I have read the journal's policy and the authors of this manuscript have the following competing interets: K.C. is a member of the scientific advisory board of Integrum Scientific, LLC. J.R.L is a consultant for Celdara Medical.

time-dependent biomarkers associated with COVID-19 severity, such as antibody titers, can substantially contribute to the development of more accurate outcome models. Here we show that models trained on immune biomarkers, longitudinally monitored throughout hospitalization, predicted mortality and were more accurate than models based on demographic and clinical data upon hospital admission. Our best-performing predictive models were based on the temporal analysis of anti-SARS-CoV-2 Spike IgG titers, white blood cell (WBC), neutrophil and lymphocyte counts. These biomarkers, together with C-reactive protein and blood urea nitrogen levels, were found to correlate with severity of disease and mortality in a time-dependent manner. Shapley additive explanations of our model revealed the higher predictive value of day post-symptom onset (PSO) as hospitalization progresses and showed how immune biomarkers contribute to predict mortality. In sum, we demonstrate that the kinetics of immune biomarkers can inform clinical models to serve as a powerful monitoring tool for predicting fatality risk in hospitalized COVID-19 patients, underscoring the importance of contextualizing clinical parameters according to their time post-symptom onset.

## Author summary

SARS-CoV-2 infected patients present with diverse clinical profiles, ranging from asymptomatic to severe respiratory failure and death. Early detection of high-risk patients is fundamental to tailor therapeutic interventions that anticipate disease progression and prevent poor outcomes. Machine learning can assist health workers in triaging patients by bringing together multiple factors, describing the patient's health, into a single model capable of predicting the most likely outcome. This can be particularly relevant in surge settings where clinical resources must be efficiently utilized. To date, most models predict COVID-19 outcomes using patient data obtained upon hospital admission. However, clinical data obtained longitudinally during hospitalization can provide a wealth of information to build more precise models. With this in mind, we monitored disease progression in 147 COVID-19 patients during hospitalization by frequently collecting clinical parameters. We show that models trained on longitudinally monitored immune biomarkers predicted mortality and were more accurate than models based on demographic and clinical data obtained upon hospital admission. Our work encourages the development of a broader computational framework that combines patient clinical data from hospital admission with longitudinally monitored biomarkers collected throughout the hospitalization to better assist health care workers with daily prognostication of COVID-19 patients.

## Introduction

Severe acute respiratory syndrome coronavirus 2 (SARS-CoV-2) is a novel human pathogenic virus that has rapidly spread worldwide to cause the devastating coronavirus (COVID-19) pandemic. The first United States (U.S) confirmed cases were reported in Washington state in January 2020 [1]. Since then, the virus has claimed the lives of 755,000 people in the U.S., including 56,300 in New York State (Nov 2021) (https://covid.cdc.gov/covid-data-tracker). In comparison to the 2003 SARS-CoV-1 outbreak, SARS-CoV-2 has a 10–20% higher infectivity

and transmissibility rate with a peak of viral shedding occurring during the asymptomatic incubation period (4–5 days), making SARS-CoV-2 considerably harder to detect and control than SARS-CoV-1 [2,3]. SARS-CoV-2 infected patients present a heterogeneous clinical profile that ranges from mild flu-like symptoms, where infection is effectively controlled, to severe respiratory failure and death, which is linked to high levels of inflammation [4]. A variety of factors are associated with severity of disease and mortality in COVID-19 patients, including demographics (e.g. age, sex), comorbidities (e.g. diabetes mellitus, hypertension, obesity), ethnicity (e.g. Black and minority ethnicity) and clinical values (e.g. body mass index (BMI) and levels of C-reactive protein (CRP) and lactate dehydrogenase (LDH)) [4–14]. Multiple studies suggest that the dysregulated immune response is one of the main factors underlying disease severity and fatality risk [2,15–25]. The immune response against severe SARS-CoV-2 infection is characterized by a combination of delayed type I/III interferon response and production of proinflammatory cytokines that recruits effector cells [2]. Patients with severe COVID-19 frequently demonstrate lymphopenia, eosinopenia and elevated levels of white blood cells (WBC) in the blood, in particular neutrophils, and differences in the humoral response [17–25]. Elevated anti-SARS-CoV-2 IgG titers correlates with length of hospitalization and is also associated with disease severity in a time dependent manner as shown by the delayed IgG response observed in deceased patients [22,23,25]. While further studies are required, current data suggests an intrinsic relationship between disease progression, severity of disease and the immune response. Therefore, characterizing the immune response over time can identify time-dependent immune features that would inform prognosis, the use of therapeutics such as COVID-19 convalescent plasma [26] or monoclonal antibodies against SARS-CoV-2 [22,27]. Most predictive models of SARS-CoV-2 infection outcomes incorporate static features, including demographics, comorbidities and single point clinical values from the hospital admission labs [8–14]. Unlike admission clinical and laboratory values, longitudinal information mirroring the evolution of disease during hospitalization might be better suited to predict disease outcome far downstream [28]. Studies on longitudinal data have mostly focused on trajectory profiling, unraveling valuable time-dependent associations between biomarkers and disease severity [29–33]. However, integration of longitudinal clinical data within a machine learning framework has rarely been reported [28,34]. Recently, Chen et al. integrated up to seven longitudinally monitored clinical parameters (LDH, lymphocytes counts, procalcitonin, D-dimer, CRP, respiratory rate and WBC counts) into a logistic regression model to predict mortality on a daily basis [28]. Model evaluation showed an area under the curve (AUC) of 75%-96% and 69%-79% when predicting mortality 1–5 days and 6–10 days before it occurs respectively. Assembly of longitudinal datasets is a challenging task that requires periodical sampling of patients over an extended period of time. Consequently, analysis of longitudinal data has only recently begun to emerge [35]. In order to bypass this limitation, mathematical representations of the immunopathology of COVID-19 and development of virtual patient cohorts were developed, elucidating potential relationships between immune response and disease severity [35,36].

Here we conducted a longitudinal study, relative to post-symptom onset (PSO), of anti-spike protein antibody levels and other immune and non-immune biomarkers in COVID-19 hospitalized patients at a single day resolution (147 patients; 1,954 blood samples spanning up to 60 days PSO) in relation to disease outcomes. We input this data into a machine learning framework to predict clinical outcomes on a day-by-day basis. Machine learning models trained on longitudinally monitored immune features outperformed models trained on static features obtained upon admission when predicting fatal outcomes. Day-to-day statistics revealed temporal associations between immune features and severity of disease. In addition, we analyzed how input features contributed to the model's prediction using Shapley value

analysis to further highlight these associations. Overall, our results suggest that longitudinal monitoring of immune markers can be a powerful tool for tracking the progress of disease course and identifying high mortality risk patients in hospitalized patients with COVID-19.

## Results

### Demographics and clinical characteristics in a hospitalized COVID-19 cohort

We conducted a longitudinal study of 147 hospitalized patients with COVID-19 at Montefiore Medical Center (MMC), located in the Bronx, New York (Fig 1A). Prior to hospitalization, the majority of patients resided at home or in nursing/rehabilitation centers from neighborhoods near MMC (Fig 1B). All 147 patients were admitted with a diagnosis of COVID-19 and had a positive real-time reverse transcription polymerase chain reaction (RT-qPCR) for

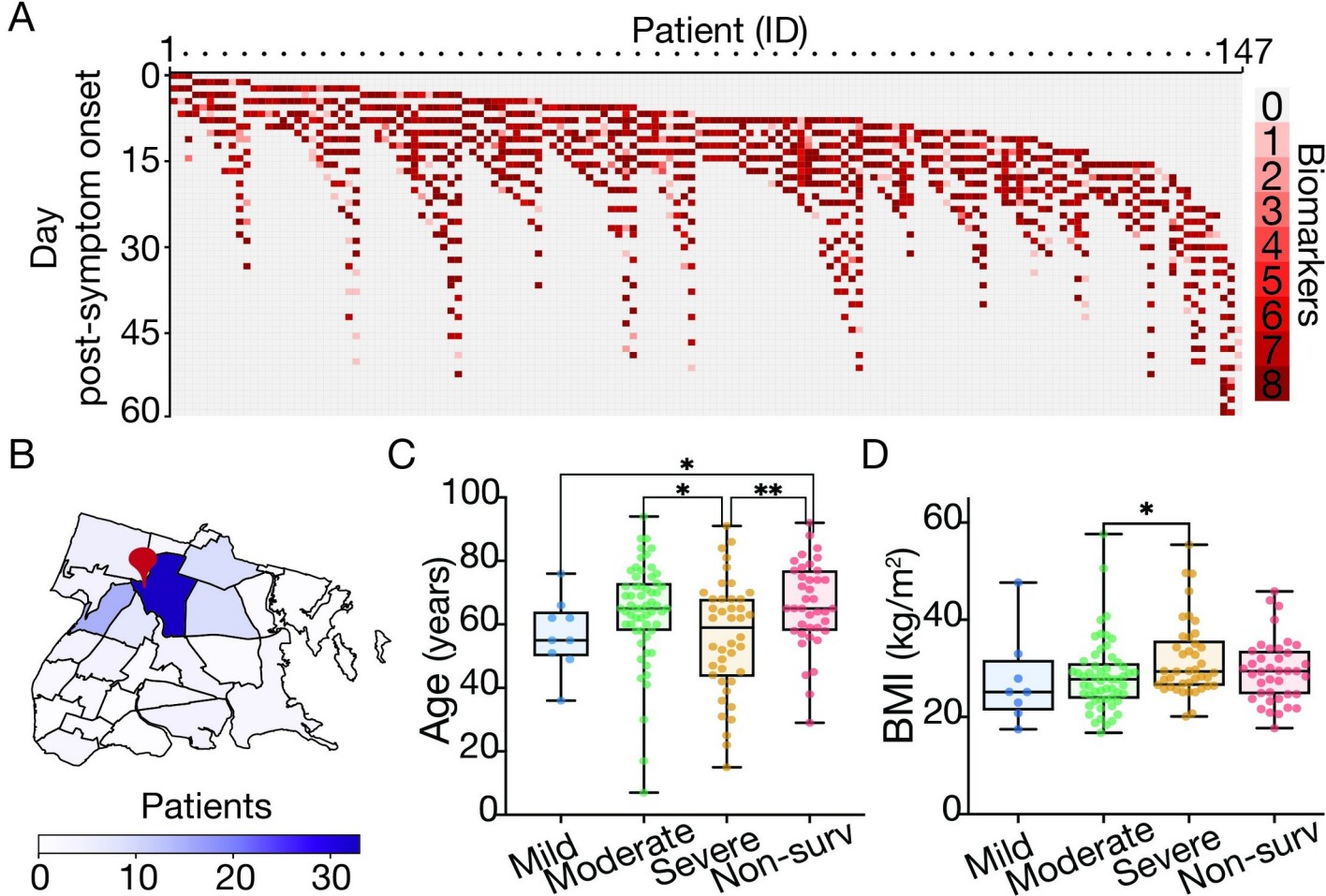

**Fig 1. Overview of patient cohort by sampling strategy, residence and demographic associations with COVID-19 disease severity.** (A) Heatmap describing the number of biomarkers (y-axis; up to eight different biomarkers, see materials and methods) measured for any given patient (x-axis; described by an internal identifier) and day PSO over course of hospitalization (patients = 147, blood samples analyzed = 1,954). (B) Choropleth map of the Bronx zip codes colored by the number of patients enrolled in the study. The red pin denotes the location of Montefiore Medical Center. (C-D) Box plot of patient's age (C) and Body Mass Index (D) by oxygen supplementation or non-survival outcome (Blue: Mild, green: moderate, orange: severe, red: non-survival). Boxes extend from the $25^{th}$ to $75^{th}$ percentiles, the whiskers represent the minimum and maximum values and the middle line corresponds to the median. Statistical significance is denoted with asterisks (Mann-Whitney; $^{*}p < 0.05$, $^{**}p < 0.01$). We define "day post-symptom onset" (PSO) as the day relative to the patient-reported onset of symptoms. We downloaded the raw data to make the map from NYC Open Data (https://data.cityofnewyork.us/City-Government/Borough-Boundaries/tqmj-j8zm) and plotted it using pandas and geopands.

SARS-CoV-2 from nasopharyngeal samples between March 1st and June 1st, 2020. Patients were admitted into the hospital at different days relative to the onset of symptoms (median admission day PSO = 7; IQR, 3–9 days) and had different lengths of stay (LOS) (median LOS = 13 days, IQR, 9–21 days). Blood samples from each patient were obtained over the course of the hospitalization. Whenever possible, up to eight different biomarkers were measured in any given blood sample (Fig 1A): anti-SARS CoV-2 spike IgG antibody titers, total WBC, lymphocyte, neutrophil, eosinophil, platelet counts, CRP and blood urea nitrogen (BUN). We numbered each day during hospitalization relative to patient-reported PSO and data were analyzed in relation to this day (e.g. when performing day-to-day statistics only patient data obtained the same day PSO were considered). Patients had a median age of 64 years (IQR, 54–72.5), median BMI of 28.4 (IQR, 25.1–33.4), with 33.6% women (Table 1). Disease severity was defined by clinical outcome and maximal level of oxygen supplementation received during hospitalization in survivors is as followed: 1) Mild: Room air (no oxygen supplementation), 2) Moderate (nasal cannula, 1–4 L/min to maintain SpO2 >92%—or non-rebreather mask), 3) Severe (non-invasive ventilation, high-flow oxygen, ≥6 L/min to maintain SpO2 >92%, or invasive mechanical ventilation 4) Non-survival (deceased during the course of hospitalization) (Table 1) (29). Individuals defined with severe illness had lower median age (59 years) relative to the moderate and non-survival groups (65 years for both groups) (Fig 1C) and a higher median BMI (29.4) than the moderate category individuals (27.8) (Fig 1D). Other common comorbidities observed in the cohort included cardiovascular disease (68.5%), diabetes mellitus (39.6%) and chronic kidney disease (26.9%) (Tables 1 and S1). Twenty six percent of patients did not survive. Median day of death from symptom onset was 21 days (IQR, 19–34 days).

## Sustained anti-spike IgG antibody response over time correlates with clinical outcome

To measure SARS-CoV-2 antibody titers we used a robust ELISA-based detection serological assay that confers high sensitivity and specificity [37] with a recombinant stabilized spike ectodomain as the antigen [38,39]. From each individual patient plasma, ELISA-dilution curves were performed from samples collected every two days during the first 10 days of hospitalization and every 3 days afterwards (Fig 2A–2C). The half maximal effective IgG concentration ($EC_{50}$) was computed from each ELISA-dilution curve and averaged on a daily basis using a rolling five-day window in order to minimize day-to-day fluctuations (see materials and methods). Then, we described the evolution of IgG titers during hospitalization as a single trajectory for each patient (Figs 2D and S1). A positive titer was defined as a $-\log_{10}(EC_{50})$ above $-\log_{10}(2.5)$ as previously reported [40]. Following this, we categorized the sustained level of SARS CoV-2 IgG for each patient into High, Medium or Low groups using the daily $EC_{50}$ values. These categories were based on the maximal and consecutive $EC_{50}$ titers (maintained for ≥ 5 days of hospitalization) and the range delimited by the 25th and 75th percentiles of all $EC_{50}$ values. Patients hospitalized for < 7 days were excluded (19 out of 147 patients; see materials and methods) due to an insufficient number of time points to categorize their sustained IgG response. Out of the 130 remaining patients, 21 patients did not have a sustained IgG response reported since their IgG titers fluctuated between the established ranges during hospitalization (intermediate category, Table 2). Patients with IgG titers > 75th percentile, for 5 days or more, were classified as High sustained IgG; and patients with maximum sustained IgG titers between the 25th of 75th percentiles or below the 25th percentile for five consecutive days or more were classified as Medium or Low sustained IgG respectively (Table 2).

Trajectory plots show that IgG titers increased within the first three weeks PSO and then remain stable throughout the length of hospitalization (Figs 2D and S1). Sustained IgG

**Table 1. Patient Characteristics.** Patients are grouped by outcome; survivors disease severity is categorized based on maximal oxygen supplementation during hospitalization. Mild: room air; Moderate: nasal cannula, or non-rebreather; Severe: high flow oxygen therapy or mechanical ventilation. BMI reported as kg/m2.a *one subject in this category was homeless.

| | All Patients | Mild | Moderate | Severe | Non-survivors |
|---|---|---|---|---|---|
| | (N = 147) | (N = 9) | (N = 59) | (N = 41) | (N = 38) |
| **Baseline and demographic** | | | | | |
| **Age**-no. (IQR) | | | | | |
| All, yrs, median (IQR) | 64 (54–73) | 55 (50–64) | 65 (58–73) | 56 (43–68) | 65 (58.7–77) |
| <40 yrs | 30 (19.5–35) | 36 (N/A) | 17 (7–30) | 30.5 (22.8–35.5) | 29 (N/A) |
| 40–64 yrs | 56 (49–61) | 53 (49.5–55) | 58 (51–62) | 52 (45.5–59) | 58 (55–61) |
| ≥65 yrs | 72 (68–78) | 64 (62–73.5) | 72 (68–78) | 70 (68–80.3) | 74.5 (68–80.3) |
| Male-no./total no. (%) | 99 (67.3) | 7 (77.8) | 37 (62.7) | 27 (65.9) | 27 (71.1) |
| BMI—median (IQR) | 28.3 (25.1–31.7) | 25.1 (21.3–30.4) | 27.7 (23.9–30.9) | 29.4 (26.4–35.7) | 29.4 (24.4–33.3) |
| **Ethnicity**-no. (%) | | | | | |
| Hispanic | 61 (41.5) | 2 (22.2) | 20 (33.9) | 22 (53.7) | 17 (44.7) |
| Not Hispanic | 75 (51.0) | 7 (77.8) | 32 (54.2) | 17 (41.5) | 19 (50) |
| Unknown | 11 (7.5) | 0 (0) | 7 (11.9) | 2 (4.9) | 2 (5.3) |
| **Race**-no. (%) | | | | | |
| Black | 53 (36.1) | 7 (77.8) | 21 (35.6) | 14 (34.1) | 11 (28.9) |
| White | 22 (15.0) | 2 (22.2) | 6 (10.2) | 8 (19.5) | 6 (15.8) |
| Other | 58 (39.5) | 0 (0) | 24 (40.7) | 17 (41.5) | 17 (44.7) |
| Unknown | 14 (9.5) | 0 (0) | 8 (13.6) | 2 (4.9) | 4 (10.5) |
| **Comorbidities**-no. (%) | | | | | |
| Hypertension | 101 (68.7) | 6 (66.7) | 41 (69.5) | 24 (58.5) | 30 (78.9) |
| Diabetes | 59 (40.1) | 4 (4404) | 23 (39) | 18 (43.9) | 14 (36.8) |
| Pulmonary Disease | 31 (21.1) | 2 (22.2) | 12 (20.3) | 7 (17.1) | 10 (23.3) |
| Heart Disease | 36 (24.5) | 2 (22.2) | 13 (22.0) | 6 (14.6) | 15 (39.5) |
| Chronic Kidney Disease | 40 (27.2) | 2 (22.2) | 17 (28.8) | 6 (14.6) | 15 (39.5) |
| Hemodialysis | 16 (10.9) | 1 (11.1) | 5 (8.5) | 1 (2.4) | 9 (23.7) |
| Immunocompromised | 26 (17.7) | 3 (33.3) | 9 (15.3) | 5 (12.2) | 9 (23.7) |
| History of smoking | 60 (40.8) | 4 (44.4) | 32 (54.2) | 10 (24.4) | 14 (36.8) |
| **Residency**-no. (%) | | | | | |
| Home | 113 (76.9) | 6 (66.7) | 44 (74.6)* | 33 (80.5) | 30 (78.9) |
| Nursing home | 34 (23.1) | 3 (33.3) | 15 (25.4) | 8 (19.5) | 8 (21.1) |
| **Length of Stay**-days | | | | | |
| Median (IQR) | 13 (9–21) | 7 (6.5–9.5) | 9 (7–15) | 19 (13–31) | 14.5 (12–24.3) |
| **Sustained IgG**-no. (%) | | | | | |
| Poor coverage | 17 (11.6) | 4 (44.4) | 11 (18.6) | 2 (4.9) | 0 (0) |
| Low | 15 (10.2) | 1 (11.1) | 5 (8.5) | 2 (4.9) | 7 (18.4) |
| Medium | 59 (40.1) | 4 (44.1) | 25 (42.4) | 13 (31.7) | 17 (44.7) |
| High | 35 (23.8) | 0 (0) | 9 (15.3) | 18 (43.9) | 8 (21.1) |
| Intermediate | 21 (14.3) | 0 (0) | 9 (15.3) | 6 (14.6) | 6 (15.8) |

response was associated with LOS, as High sustained IgG group tend to stay longer in hospital (Kruskal-Wallis p = 1.2 x10$^{-3}$, Table 2). Unlike patients with a sustained Medium and High IgG response, patients with a Low sustained IgG response had negative or very low levels of IgG against the spike and nucleocapsid proteins (S2 Fig). Half (50.8%) of the 130 patients mounted a Medium IgG response, plateauing after 15 days PSO and 34.6% of the cohort had a High sustained IgG titer.

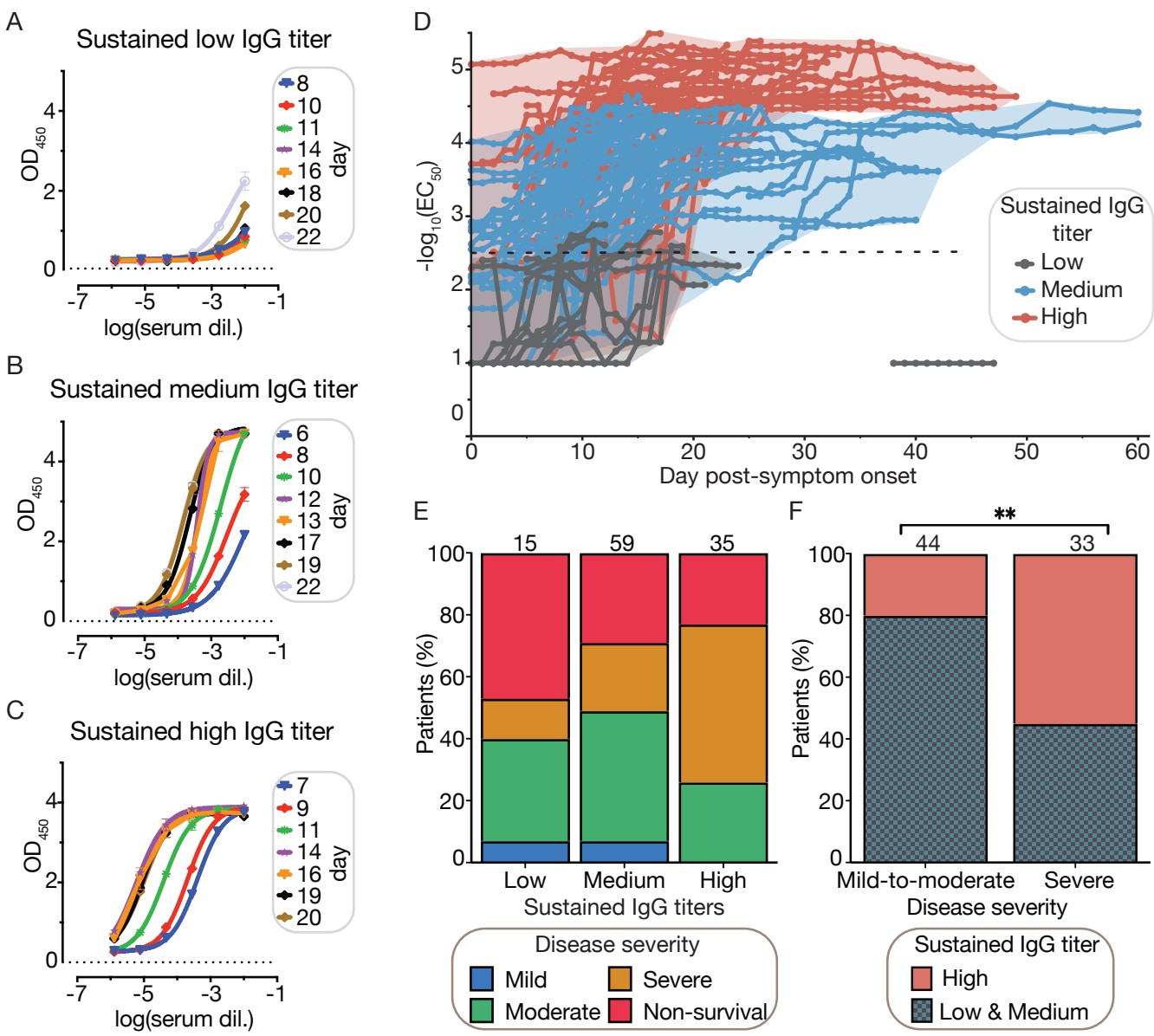

**Fig 2. Sustained IgG titers are associated with severity of disease.** (A-C) SARS-CoV-2 spike IgG ELISA titration of plasma samples over the course of hospitalization for representative patients showing three distinct evolutions of IgG titers. ELISA experiments were carried out as two experimental replicates, each consisting of two technical replicates. (A) Representative patient with a sustained low IgG titer during hospitalization. (B) Representative patient with a sustained medium IgG titer during hospitalization. (C) Representative patient with a sustained high IgG titer during hospitalization. (D) $EC_{50}$ titers over time were used to categorize patients with $\geq 7$ days of hospitalization (n = 130) into three categories that describe the sustained IgG titer: i) low (grey), medium (blue) and high (red). Categorized patients are required to show daily IgG $EC_{50}$ titers for at least for 5 consecutive days within the range delimited by the 25th and 75th $EC_{50}$ percentiles (sustained low IgG: $-\log_{10}(EC_{50}) \leq$ 25th perc; sustained medium IgG: 25th perc $> -\log_{10}(EC_{50}) \leq$ 75th perc; and sustained high IgG: $-\log_{10}(EC_{50}) >$ 75th perc; see materials and methods and S1 Fig). (A-D) We define "day post-symptom onset" (PSO) as the day relative to the patient-reported onset of symptoms. (E) Mosaic plot describing the distribution of severity of COVID-19 relative to the sustained IgG titer class. (F) Distribution of sustained IgG titers relative to the severity of disease among survivors (77 patients), patients with mild and moderate disease were merged into a single category (Chi-square, **p < 0.01).

We did not find a significant association between the category of sustained IgG response and mortality (Chi-square p = 0.24). However, the lack of significance could be influenced by the small dataset (only 15 hospitalized patients had a Low sustained IgG response, where

**Table 2. Length of hospitalization is associated with sustained IgG titer against SARS-CoV-2 spike protein.** 130 patients out of 147 (patients with less than 7 days of hospitalization were excluded) were categorized into three separate categories ($<25^{th}$, $25^{th}$-$75^{th}$ and $>75^{th}$ percentiles of all $EC_{50}$ values) according to their corresponding sustained IgG response, which was maintained for at least 5 consecutive days during hospitalization. Out of the 130 patients, 21 patients with IgG titer fluctuating between the three established categories were separated into an intermediate category. Kruskal-Wallis test shows a significant association between the LOS and sustained IgG categories (p value = $1.2 \times 10^{-3}$).

| Sustained IgG titer | Total IgG | Low IgG | Medium IgG | High IgG | Intermediate category |
|---|---|---|---|---|---|
| Number of patients (%) | 130 | 15 (11.5) | 59 (45.3) | 35 (26.9) | 21 (16.1) |
| Length of stay, days median (IQR) | 15 (9.8–23) | 9.0 (8.0–12.0) | 14.9 (9.0–20.0) | 16.0 (13.0–24.0) | 20.0 (10.0-28-5) |

46.7% of them died, Fig 2E). In contrast, we found a strong association between severity of disease and the sustained IgG response categories (Chi-square square p<0.01) among survivors: only 20% of patients with mild-to-moderate disease show a sustained High IgG response, as compared to 55% of patients with severe disease (Fig 2F).

## Trajectory analyses reveal different IgG kinetics between survivors and non-survivors

Sustained Low, Medium and High IgG response categories describe the maximal IgG titers over the entire length of hospitalization but do not provide information about the kinetics of the IgG response. Next, we tested the association of admission IgG titers and the kinetics of IgG response with disease severity (mild, moderate, severe and non-survival).

**Low IgG titers early PSO and high IgG titers later PSO are associated with more severe disease.** There was no association of hospital admission IgG titers or serostatus with disease severity (S3 Fig). However, when comparing median IgG titers relative to the onset of symptoms we observed two time windows where IgG titers are associated with disease severity (Fig 3A). First, early PSO (days 2–11) non-survivors have lower IgG titers than survivors (Mann-Whitney p < 0.05; Figs 3A and 3B and S4). Survivors showed an early median IgG titer near or above the seroconversion threshold (-$\log_{10}(EC_{50})$ = 2.5) during the first week PSO whereas most non-survivors seroconverted during the second week (Fig 3A). Later PSO, patients with severe disease and non-survivors had higher median IgG titer than patients with moderate disease severity (Fig 3A). Day-to-day statistics on patients grouped by maximal oxygenation supplementation shows that during this late time window (comprising days 18 to 23) patients receiving oxygenation via nasal cannula consistently showed lower IgG titers than non-survivors and patients with more severe disease (Figs 3C and S5).

**Non-survivors have a delayed IgG response PSO.** Patients with mild disease severity showed minimal variation in titers throughout hospitalization with a median IgG titer at or above the positive threshold during the 1st week PSO (Fig 3A). In contrast, IgG titers increased between the 1st and 3rd week PSO in patients with moderate and severe disease and non-survivors. When comparing the date at which the maximal titers are reached we observed that non-survivors tend to reach a plateau later than survivors (IQR 9–16 days and 12–19 days respectively; Mann-Whitney p = $1.2 \times 10^{-2}$, Figs 3A and S6). Thus, survivors were more likely to have higher titers, be seropositive and reach a plateau early PSO compared to non survivors.

## IgG titers are associated with ethnicity

We determined if there was a temporal association between IgG titers and race and ethnicity over time. Although patients of Black race tend to reach maximum IgG titers a few days earlier than White patients (S7A Fig, IQR 9–14 days and 10–20 days respectively), those differences were not statistically significant (Mann-Whitney p = 0.14). Similarly, we found no significant association between severity of the disease (or sustained IgG response) and race

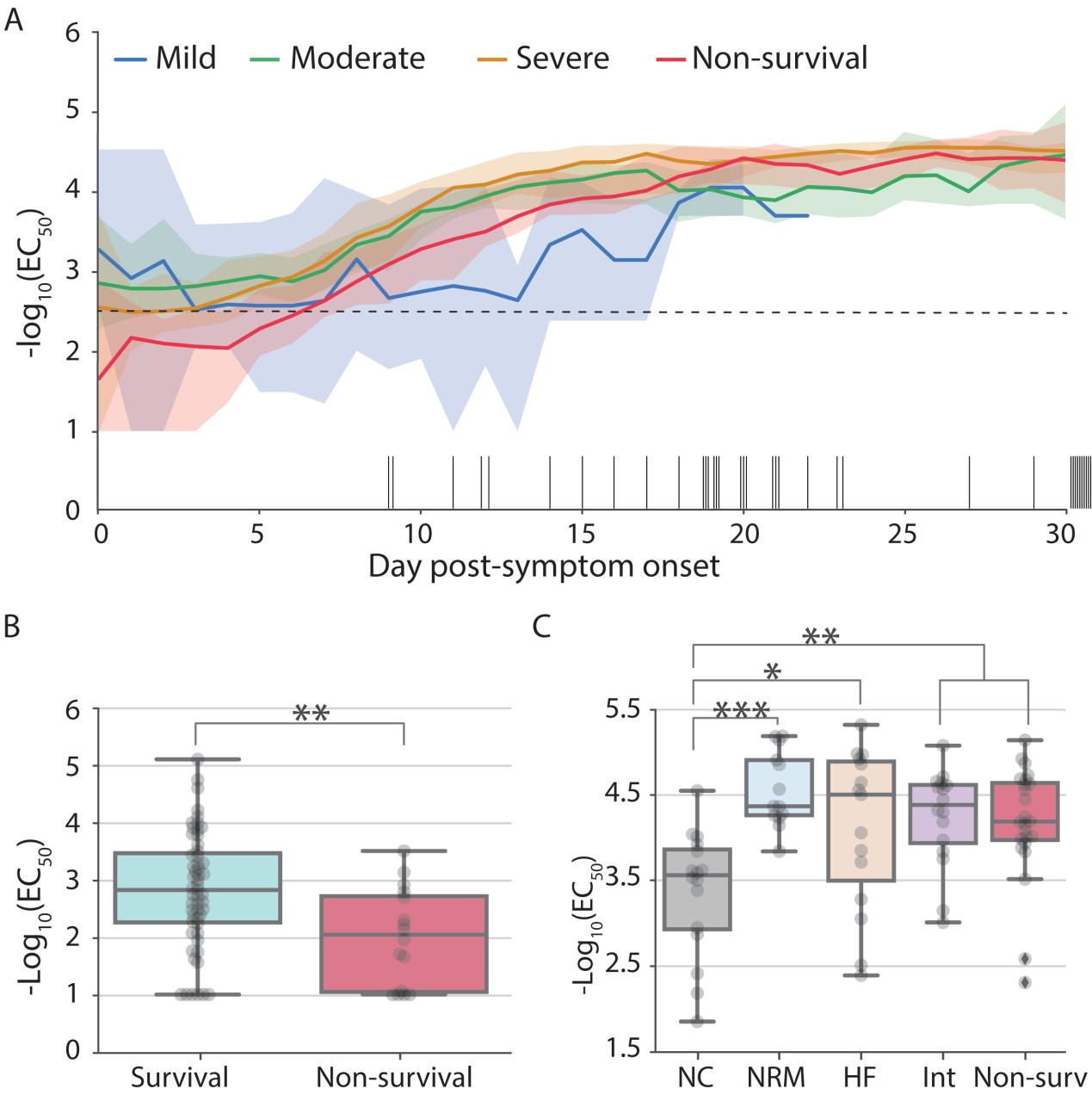

**Fig 3. IgG titers are associated with mortality early post-symptom onset and with severity of disease later during hospitalization.** (A) Median IgG titer at a given day during hospitalization by disease severity: mild (blue), moderate (green), severe (orange) and non-survivors (red). Back bars on the x-axis describe the day of death for each deceased patient. For clarity purposes, only the first 30 days of hospitalization are shown (blood samples were collected up to day 60 PSO). Shaded areas correspond to the 90% confidence intervals. IgG positivity threshold is indicated with a horizontal black dotted line at -$\log_{10}(EC_{50})$ = 2.5. We define "day post-symptom onset" (PSO) as the day relative to the patient-reported onset of symptoms. (B) IgG titers for survivors (cyan) and non-survivors (red) at day 4 PSO (used here to represent the statistical differences observed between convalescent and deceased patients from days 3 to 8). (C) Box-plot at day 18 PSO (representative day of the statistical differences observed from days 18 to 24) describing IgG titers (-$\log_{10} EC_{50}$) by patient outcome: survival patients requiring oxygen supplementation (nasal canula: grey, non-rebreather mask: blue; high-flow: salmon; intubation: pink) and non-survival patients (red). Boxes extend from the 25th to 75th percentiles, whiskers extend to the lowest and highest data point within 1.5 interquartile range of the lower and upper quartiles, the middle line corresponds to the median. Statistical significance is denoted with asterisks (Mann-Whitney; *p < 0.05, **p < 0.01).

(S7B and S7C Fig). Hispanic and non-Hispanic subjects had a median IgG titer above the sero-positive threshold early PSO and their IgG titers steadily increased during the second week to reach a plateau at days 10–17 (S8A Fig). Hispanic patients had higher median IgG titer than non-Hispanics from days 19–40 PSO (Mann-Whitney $p < 5 \times 10^{-2}$; S8A Fig and S2 File). In line with this observation and relationship between IgG titers and severity of disease later PSO (see above), we found an association between mild-to-moderate and severe outcomes and ethnicity, where 51% of Hispanic survivors suffered from severe disease as compared to 30% of non-Hispanic survivors (Chi-square p = 0.03) (S8B Fig). In addition, there was a similar trend when comparing sustained IgG responses, with 47% and 26% of Hispanic and non-Hispanics with a sustained high IgG titer (Chi-square p = 0.03; Supp 8C). Our data suggests that ethnicity is associated with IgG titers and severity of symptoms but not mortality in the late window PSO.

## Trajectories of specific biomarkers are associated with COVID-19 mortality

Anti-spike IgG antibody is only one facet of COVID-19 response; other immune and inflammatory features have been shown to be associated with disease outcome [41]. Similar to the temporal analysis carried out on IgG titers, we compared the distribution of WBC, neutrophils, lymphocytes, eosinophils, platelets, CRP and BUN on a five-day rolling window basis across our disease severity categories (Fig 4). Our results show that non-survivors have a distinct profile from the survivors (mild, moderate or severe disease). As a group, non-survivors showed higher levels of WBC, neutrophils, CRP and BUN and lower levels of lymphocytes and eosinophils than patients who survived. However, we did not observe a clear pattern distinguishing patients with mild, moderate or severe illness. WBC and neutrophil levels were statistically higher for non-survivors between days 7 to 32–33 than for survivors (Mann-Whiney $p < 1 \times 10^{-3}$, Fig 4A and 4B and S1 File). BUN levels in non-survivors remained significantly higher from day 6 after PSO until day 43 (Fig 4H and S1 File). CRP levels abruptly increased over 10 mg/dl within the first ~4 days in the non survivors and remained significantly higher from days 5 to 28 (Fig 4G). On the contrary, lymphocyte levels showed minimal variation throughout hospitalization and the non-survivors displayed consistently lower levels relative to the survivor group from days 2 to 35 (Fig 4C and S1 File).

## Time-dependent clinical features outperform static features obtained upon admission to predict mortality

Demographic, clinical and/or biochemical features extracted upon admission have been used extensively to predict fatality risk in COVID-19 patients [41]. However, there is a need for orthogonal approaches that leverage longitudinal information to inform the early assessment of fatality risk in patients that do not yet show severe signs of disease [28]. In order to compare the predictive value of longitudinal features and static features, we first generated models based on patient data taken at the time of admission to predict severity of disease and mortality. We next utilized longitudinal data collected during the patient's hospital stay focusing on immune biomarkers to predict mortality and intubation in near real time (Methodology detail in S3 File).

**Patient electronic-medical record data is modestly predictive for COVID-19 outcomes.** To identify biomarkers that could improve an initial clinical assessment of patient prognosis we implemented machine learning models to predict severity of the disease using only information that would be available upon admission, including immune, clinical and demographic features (e.g. neutrophil counts, LDH levels or age; S10 Fig). We compared

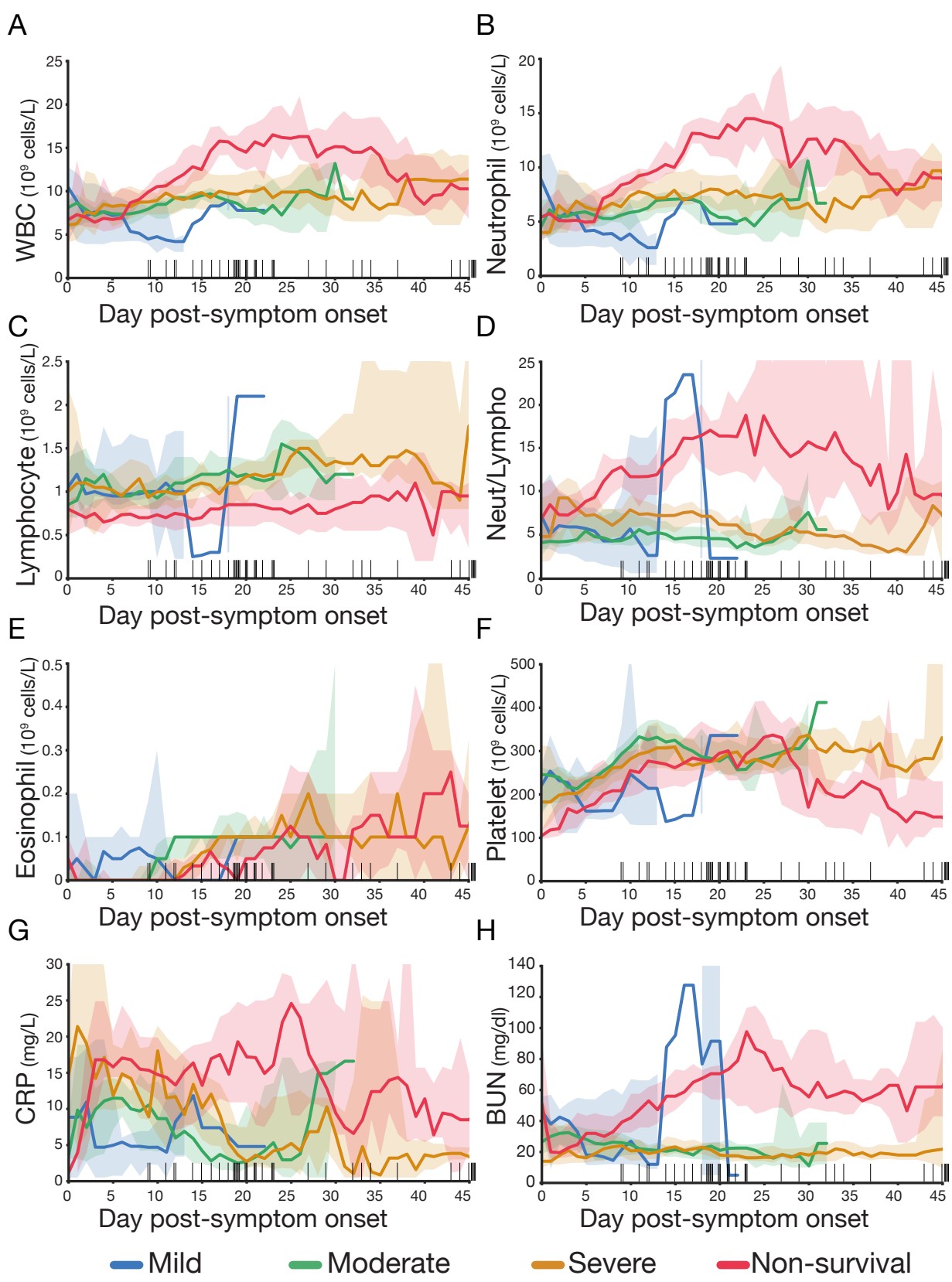

**Fig 4. Dynamic variation of leucocytes, inflammatory biomarkers and renal function during hospitalization is associated with mortality.**
Levels of (A) total white blood cell count (WBC), (B) neutrophils, (C) lymphocytes, (D) neutrophil to lymphocyte ratio, (E) eosinophil, (F) platelet, (G) CRP and (H) BUN during hospitalization. Patients are categorized according to the severity of disease: mild–blue, moderate—green, severe—orange and non-survival—red. Cell count and CRP and BUN levels are averaged using a five-day sliding window. Back bars on the x-axis are the day of death for each deceased patient. Shaded areas correspond to 90% confidence intervals. The group size of the mild category is reduced to one to three patients after day 11 PSO. We define "day post-symptom onset" (PSO) as the day relative to the patient-reported onset of symptoms.

logistic regression, random forest, and a two-layer perceptron network and found that all models performed similarly based on area under the receiver operating characteristic (AUROC) and precision-recall curve analysis (Figs 5A and S9). In general, our predictive performance was better for more severe outcomes; we achieved 63.2% and 66.0% AUROC for non-survivors and patients with severe disease respectively, compared with AUROC of 46.1% and 56.0% for patients with mild and moderate disease respectively. Next, we identified the features which most strongly contributed to the model's predictions. For the best performing model, the random forest, we computed feature importance scores to describe the weight each feature has towards the final prediction (S10 Fig). We found six features that are highly relevant for classification (features among the top 30% used to predict at least three categories): LDH, neutrophils, platelets, BMI, BUN and initial temperature [42–46]. Other features known to be associated to COVID-19 severity and mortality are also predictors of specific categories in our cohort [47–49]. For instance, age appears among the top features when predicting mild and severe disease, while levels of albumin and lymphocytes predict severe disease and mortality. We are also able to distinguish features that do not contribute to clinical predictions (features among the bottom 30% used to predict at least three categories): sore throat, residing at assisted living facility, diarrhea, sex, or Black race. In contrast to previous findings, we observed a negligible contribution of admission eosinophil counts in our model [50].

**Daily immune biomarkers can predict COVID-19 events on a day-to-day basis.** We next sought to integrate the findings that longitudinal immune biomarkers are associated with outcomes (mortality and intubation) into a predictive model. We hypothesized that longitudinal biomarkers are more suited for predicting outcomes within the next few days rather than the patients' final outcome. To address this question, we implemented machine learning models to predict a patients' mortality status (Fig 5B and 5D) and intubation (S11 Fig) within the next k days using immune biomarkers that would have been available on a given day. In a representative example, for k = 5, we were able to achieve a mean AUC score of 70.2% when predicting death with a random forest that used the daily values of lymphocytes, WBC counts, IgG titers, and neutrophils along with the day (Fig 5B), and we found similar results as we varied k from 2 to 10 (Fig 5C and 5D). When predicting intubation, the performance of the classifier decreases (S11C and S11D Fig). On precision-recall, intubation appears to perform better, but this outcome is due to the higher prevalence of intubation-positive samples in the dataset (39%) relative to death (12.7%), which makes high levels of precision more difficult to achieve (S11B–S11D Fig). Additionally, we found that the performance of the longitudinal classifier to predict mortality varies in a time-dependent manner; classifier performance improves over time until reaching a peak in the firth week PSO (Fig 5B–5D). When evaluated on a weekly basis (k = 5), the mean AUROC increases from 60.2% in the first week of evaluation (PSO) to 80.2% in the fifth week of evaluation (PSO) (Fig 5B). The predictions were not as accurate for longitudinal prediction of intubation; AUROCs were 53.8%, 56.2%, 55% and 58.9% for weeks 1–4 respectively (S11E Fig).

**Shapley values on the longitudinal model identifies patient clusters with higher likelihood of outcomes.** To understand how these immune biomarkers relate to the likelihood of mortality, we estimated the Shapley values associated with the features of our longitudinal

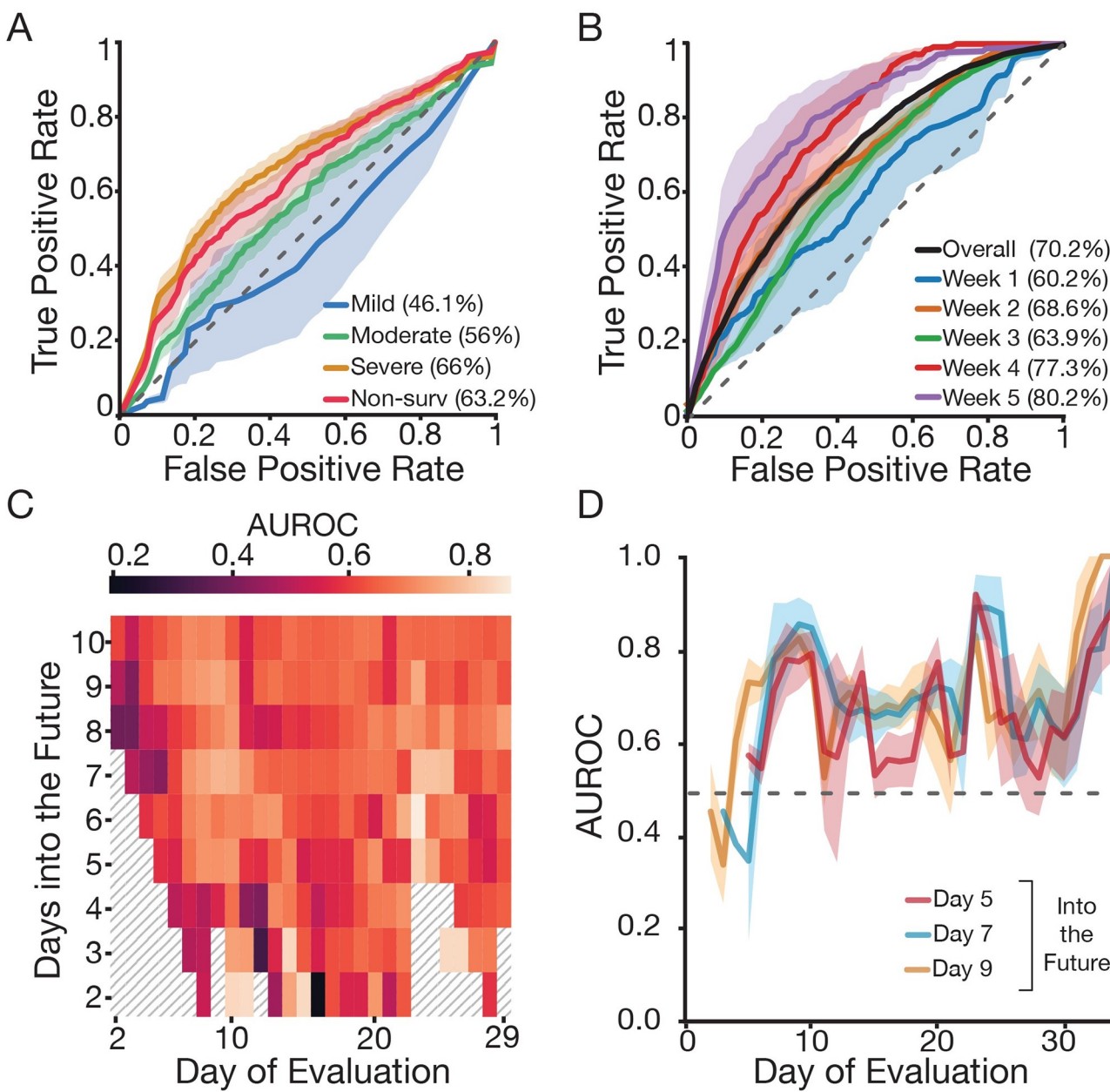

**Fig 5. Time-dependent clinical and laboratory data outperform day of admission data to predict fatal outcome.** (A) ROC performance on predicting severity of disease and mortality using a random forest classifier based on data from the EMR (including clinical, laboratory data and demographics; see S7 Fig) on admission. Shaded areas correspond to ± 2 standard error of the mean. (B) ROC performance on predicting mortality within the next five days (k = 5) using a neural network based on time dependent clinical features (IgG titers, total white blood cell count, neutrophils, lymphocytes, eosinophils, platelets, CRP and BUN; see Fig 4). ROC curves correspond to the overall performance (purple) as well as weekly predictions (relative to the patient-reported onset of symptoms) during the length of hospitalization. A-B) Legend describes the corresponding area under the curves. (C) Heatmap shows AUROC for daily mortality classifier (neural network) for different values of k (number of days into the future to predict; y-axis) and day of evaluation (restrict test set to a given day). (D) AUROC as a function of day of evaluation for various values of k. Regardless of k, the performance of the classifier remains consistent and improves over the course of a month. (C-D) We define "day of evaluation" as the day relative to the patient-reported onset of symptoms when mortality risk was assessed. ROC: receiver operating characteristic; EMR: electronic medical record; AUROC: area under the receiver operating characteristic.

model relative to the prediction. Shapley values assign an attribution to features on a per-sample basis, based on how they contribute to the prediction, and enabling us to understand how the predictive value of a feature changes with that feature's value. We plotted the mean Shapely values for each sample compared to their lab values (Figs 6A–6D and S12 and S13 and S4 File). For IgG titers, we observed a quadratic relationship where both high and low values were associated with a higher propensity of fatal outcomes relative to a middle value (Fig 6A). Such relationship describes a gradual shift from negative to positive as the number of days increases (S12 Fig): lower IgG titers are more informative when predicting mortality early PSO and higher IgG titers are more informative when predicting mortality later PSO. This observation agrees with the associations we detected in regard to IgG titers being non-linearly correlated with outcome, with both high and low values of IgG associated with worse outcomes in a time-dependent manner. In addition, our models associated a higher attribution to increasing values of WBCs, neutrophils and day (PSO) of measurement, while assigning lower attributions to higher levels of lymphocytes (Figs 6B–6D and S13), in agreement with our statistical analysis described above (Fig 4).

Next, similar samples were grouped together based on their attributions to identify and characterize groups of samples with higher or lower likelihood to be associated with mortality (Fig 6E). Thereby, we identified 8 clusters with varying likelihoods of mortality during the following five days. Enrichment analysis on each cluster revealed that clusters 3 and 4 were significantly enriched for patients who survived while cluster 2 was enriched for non-survivors (cluster 6 was also enriched for non-survivors but became non-significant after correcting for multiple hypothesis) (Fig 6E). We observed a distinct distribution of immune biomarkers when comparing clusters (S14 Fig) that aligns well with the associations described above (Figs 3 and 4). For instance, patient samples in cluster 3 (enriched for survivors) show the lowest levels of WBC and neutrophil counts while patient samples in cluster 4 (also enriched for survivors) show the highest values of lymphocyte counts

## Discussion

Most current models to predict COVID-19 disease severity rely on invariable features obtained upon hospitalization [7–14,28,50]. However, time-dependent measurements monitoring disease progression can flag high risk patients by detecting changes in their lab values and identify the optimal therapeutic windows for administration of a limited arsenal of anti-inflammatory drugs and antibody-based therapies. Since the immune response is associated with COVID-19 severity, longitudinally monitored immune biomarkers can provide a precise description of the disease trajectory [2,15–25]. The work described here followed the evolution of anti-SARS-CoV-2 spike IgG, immune cells and other non-immunological biomarkers at single-day resolution throughout the length of hospitalization on a cohort of 147 COVID-19 patients. This enabled us to capture a detailed and unique time frame progression of the disease and clinical features associated with different outcomes. Our analysis of IgG trajectories suggests that a potent, early immune response is associated with survival in hospitalized patients. First, survivors had a higher basal IgG titer (at or above the seropositive threshold) early PSO and a faster seroconversion. Secondly, patients who died had much lower IgG titers from days 2–11 compared to patients who survived, highlighting a poor or delayed IgG response early in the illness. Similar observations have been reported for non-survivors where robust and early IgA and IgM responses were detected, but delayed and incomplete IgG immune responses were also observed pointing to a lack of early IgG class switching [22,23,25]. We also observed key differences during the later phase of hospitalization (days 18–23 PSO) where patients with higher anti-spike IgG antibody levels required a higher level of oxygen supplementation. This

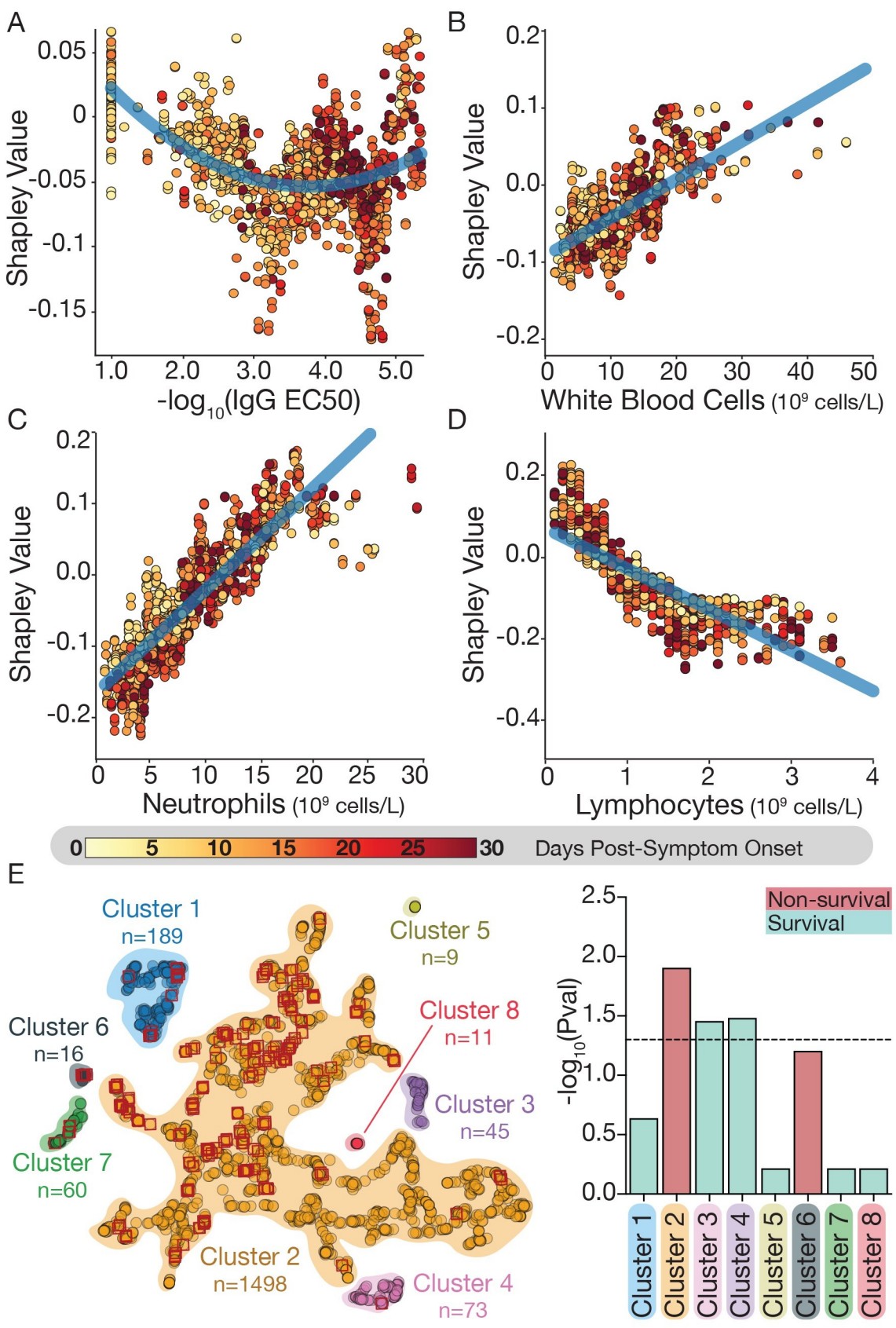

**Fig 6. Shapley value analysis reveals how immune biomarkers relate to mortality.** (A-D) For each of 4 immune biomarkers, the mean Shapley value compared to the biomarker value was plotted for each sample in the dataset (see materials and methods for details). The higher the Shapley value, the more the variable is predictive for mortality (and vice-versa): (A) IgG titers; (B) WBC count; (C) neutrophil count; (D) lymphocyte counts. Shapley values are colored based on the corresponding day PSO. (E) UMAP clustering of mean Shapley values. Red diamonds correspond to samples from patients that died within five days (k = 5) of the evaluation day. Right panel describes the significance of the enrichment for survivors (cyan) or non-survivors (red) in each of the obtained clusters using the hypergeometric test corrected for multiple hypothesis testing by Benjamini/Hochberg. Dotted horizontal line corresponds to the significance level at p = 0.05.PSO: post symptom onset.

observation aligns with previously reported findings where the titers of total Abs, nAbs, anti-RBD IgG and anti-NP IgG and IgM were found to correlate with the severity of the disease [51–54]. In addition, the sustained high titer IgG response observed for a subset of hospitalized patients with severe disease is similar to that described in acute viral infections such as those caused by Ebola virus and Dengue virus, and chronic inflammatory conditions such as systemic lupus erythematosus [55–57]. It remains unclear why the Ab response correlates with severity of the disease later during hospitalization; one possibility is that the quality of Ab produced is inadequate in controlling the disease and could even augment pathogenicity. Beside neutralization properties, antibodies are also involved in a plethora of effector functions via their Fc domain. Recent studies have found that patients with severe COVID-19 have a higher proportion of afucosylated Fc IgG, which is known to trigger a highly inflammatory response via FcγR pathways [58]. Overall, these results illustrate the time-dependent association of anti-SARS-CoV-2 S IgG titers with different COVID-19 outcomes: high titers associate with survival during an "early time window" (days 2–11 PSO), whereas during a "late time window" (days 18–23 PSO) high titers associate with more severe illness. In addition to IgG titers, we also found temporal associations between other immune and clinical biomarkers and mortality. Non-survivors experienced i) higher levels of WBC (neutrophils), CRP and BUN during the first week PSO until weeks 4–5 and; ii) consistently lower lymphocyte levels throughout the hospitalization period. Altogether, these findings suggest that immune biomarkers, measured at different time points, might have prognostic implications, not only by identifying high risk patients but also by predicting fatal outcomes in the near future. To this end, we evaluated classifiers separately trained on static variables obtained upon hospital admission and immune biomarkers longitudinally monitored during hospitalization. Our classifier trained solely on demographic factors alongside lab values collected at day of admission is useful for identifying high risk patients (AUROC = 63.2%). We found six features that are highly relevant for classification: LDH, neutrophils, platelets, BMI, BUN and initial temperature. These features align with our temporal statistical analysis, for instance neutrophil and BUN levels are significantly higher among non-survivors after the first week of hospitalization. Moreover, other studies also describe the same features as potential prognostic biomarkers for COVID-19 severity and/or mortality, validating our approach [14,42–46]. These observations are also in line with other studies which have produced similar models [59,60]. For example, Knight et al. [11] developed a model for predicting COVID-19 mortality from eight admissions features. Their model showed a higher performance (AUROC = 79%) but did so using a much larger dataset (35,463 vs 147 patients). We demonstrate that, in addition to static features obtained upon hospital admission, longitudinally monitored immune biomarkers can also be leveraged to predict fatal outcome in COVID-19 hospitalized patients. The classifier trained on time-dependent immune features can predict not just the patient's likelihood of mortality, but also when that may happen. This model has a reasonable overall AUROC of 70.2% which increases, as the disease progresses, up to 80.2% in the fifth week PSO. Our observation that AUROC increases over time suggests that lab values become more informative later during hospitalization. In addition, we also observed a positive correlation between day PSO and the

likelihood of fatal outcome. Our model did not predict intubation as accurately as mortality, potentially reflecting that patients with severe COVID-19 and poor prognosis were provided palliative care rather than intubation (55).

We provide novel insights into patient outcomes by interpreting the features that drive our model using Shapley value analysis and identifying how the contributions of different features to our models change as the values of those features evolve. Comparing Shapley values and raw lab values unveiled significant trends which align with and reinforce previous findings. For example, low IgG titers are more informative during the early time window to predict mortality (while high IgG titers are more informative days later PSO) and higher WBC and neutrophil counts are more predictive for fatal outcomes (Figs 6 and S12) [18–20,61,62]. In addition, low lymphocyte levels are predictive for mortality, and in fact many non-survivors had lymphopenia (lymphocytes $< 1x10^9$ cells/L) throughout their hospitalization. Eosinopenia has also been reported as a biomarker of more severe outcomes, but our results did not support this finding [62]. We found that eosinophil counts contributed little to predictions when included in the model, and we only found significant differences in their value between outcomes from days 8–11 [20]. Incorporating interpretability mechanisms into clinical models can help identify biases and incorrect predictions due to differences between the training and real data [63–67]. Using Shapley values and feature importance scores, we showed that our models are consistent with both our statistical observations and with previous observed relationships in the literature. Our models consistently assign similar attributions to similar features across cross-validation runs, and those attributions align with expectations of how these immune biomarkers should correlate with outcomes. Thus, we can be confident that our models are detecting meaningful patterns in patient data.

Over the course of the pandemic, clinical practice has evolved with the use of antivirals, monoclonal antibodies and steroids to treat COVID-19 [68]. As treatment of COVID-19 changes, additional biomarkers to predict outcome may need to be investigated. Our study did not capture SARS-CoV-2 viral loads, which have been associated with clinical outcomes in other studies and would likely further inform our predictive models of infection outcomes [69,70]. Also capturing and analyzing day to day changes in oxygen supplementation and additional clinical and laboratory information (e.g. additional Ig titers) could provide more granular information to further inform pathogenesis models and treatment of severe COVID-19 [71]. Here we present a framework which could be utilized in future studies. An additional limitation of this study is the size of our cohort (N = 147) and its bias towards hospitalized patients. Since the main objective of our study is to evaluate the potential to predict fatal outcomes using only longitudinally monitored biomarkers, we opted for a detailed characterization of the biomarkers trajectories throughout the hospitalization stay. Although our dataset represents the hospitalized spectrum of COVID-19 disease, it would be interesting to also study outpatients, longitudinally monitored in a similar fashion. A multi-center re-evaluation of the proposed longitudinal model, using a more extensive cohort, would be necessary prior to incorporating this model into the clinic. Nevertheless, the reported time-dependent associations of longitudinally monitored biomarkers can still be useful while assessing illness evolution.

The extent and severity of this pandemic has challenged healthcare providers to find better ways to care for a surge of patients and to identify those who are at higher risk for poor outcomes. Current models, trained on static features upon hospital admission and larger cohorts, have shown respectable-to-excellent results in predicting mortality (AUCs ranging from 68 to 98%, depending on the features, models and datasets used) [9,10]. These, however, do not consider features that describe how disease progresses over time during hospitalization, overlooking a wealth of clinical data to build more precise models. Here, we identified a set of immune biomarkers which can be used to longitudinally monitor the time-course of hospitalized

COVID-19 patients. Our time-dependent models, inherently different from most current models, show that frequent testing of specific immune biomarkers could be used to preemptively detect COVID-19 fatalities and estimate when that may occur. There is a broader interest in real-time or near-real-time monitoring of patient biometrics, and our model represents one example of this trend applied to a particularly urgent challenge. The results presented here encourage the development of a broader computational framework that combines static features (including demographics, comorbidities and other parameters obtained upon hospital admission) and longitudinally monitored clinical features collected during hospitalization to assist in the daily prognosis of COVID-19 patients and tailor therapeutic interventions at particular time-frames.

## Materials and methods

### Ethics statement

Study was approved by the Institutional Review Boards (IRB) of the Albert Einstein College of Medicine (2016–6137), and informed consent was waived because the study was retrospective, and involved no more than minimal risk to subjects.

### Study design and sample collection

This study was conducted at Montefiore Medical Center (MMC) in association with the Albert Einstein College of Medicine and was conducted with approval of Albert Einstein College of Medicine Institutional Review Board. The study included all patients who required MMC hospital admission with COVID-19 who had a positive RT-PCR test for SARS-CoV-2 infection between March 1, 2020 through June 1, 2020. Patients were monitored daily and blood samples were collected to measure serum reactivity (to the SARS-CoV-2 spike protein) repeatedly over the course of hospitalization. 147 patients were monitored by days PSO, ranging from 1–60 days PSO. Clinical and laboratory data was extracted from electronic health records.

Demographics including race and ethnicity, age, gender, living residence status, and BMI and comorbidities, including smoking status were recorded. Vitals and oxygen supplementation status were recorded for each patient throughout their hospitalization. Presenting vitals and presenting symptoms are listed in S1 Table. Maximum temperatures were collected, and if temperatures over 100.4° F were recorded, the number of days with fever was noted. The maximum oxygen supplementation received was recorded as a surrogate for lung function and representation of disease severity. Patients were divided into four categories—mild, moderate, severe, and non-survival. The mild category included patients who did not require supplemental oxygen. The moderate category had patients who required nasal cannula (1-4L/min to maintain SpO2 >92%) or non-rebreather mask. The severe category included patients that required non-invasive ventilation, high-flow oxygen (≥6L/min to maintain SpO2 >92%) or invasive mechanical ventilation or extracorporeal membrane oxygenation (ECMO). Non-survivors died during the hospitalization. Lab values that were monitored included: complete blood counts, liver function tests, BUN, CRP, D-dimer, and coagulation panel.

### Sample collection, handling and longitudinally monitored biomarkers

Daily blood samples were obtained by venipuncture (BD Vacutainer, Serum), for clinical care and serum was obtained by centrifugation of remnant clinical samples which, aliquoted and stored at -80°C. Rarely when serum was not available, plasma was collected. Prior to the measurement of anti-spike IgG antibody titers, samples were heat-inactivated for 30 minutes at 56°C and stored at 4°C. Samples were handled under BSL-2 containment in accordance with a

protocol approved by the Einstein institutional biosafety committee. Daily samples were used to measure i) anti-SARS-CoV-2 spike IgG antibody titers, ii) WBC counts, iii) lymphocyte counts, iv) neutrophil counts, v) eosinophil counts, vi) platelet count, vii) CRP, and viii) BUN. Whenever possible, all eight biomarkers were measured in each sample.

## SARS-CoV-2 spike specific IgG robotic ELISA

Half-area microtiter ELISA plates (Corning #3690) were coated with 25 µl of 2 µg/ml purified spike protein in phosphate-buffered saline (PBS) overnight at 4˚C [38]. Plates were then washed with PBS-T (PBS, pH 7.4 + 0.1% (v/v) Tween) before being blocked for 1 h at 25˚C with PBS-T + 3% (v/v) milk (Bio-Rad #170–6404). Serum was serially diluted starting from 1:100 to 1:777600 (6 points curve) in V bottom 96-well source dilution plates (Axygen #P-96-450V-C-S) with PBS-T. From this point, the APE Elite automated ELISA system (DAS, Italy, www.dasitaly.com) was used for the remaining ELISA steps. After PBS-T wash of the blocked plates, an APE Elite reference baseline (wavelength 630 nm) was performed. Diluted samples were added to well in duplicate using 2 integrated steel needles. Plates were incubated for 1 h at 37˚C with gentle shaking before being washed with PBS-T. Secondary antibody (1:3,000 in 1% milk PBS-T): anti-human IgG-Horseradish peroxidase (HRP) (Thermo Fisher #31410) was added and incubated for 1 h at room temperature. Plates were washed prior to development with ultra-TMB ELISA substrate solution at room temp (Thermo Scientific #34029). Plates were incubated for 5 min before quenching the reaction with 0.5 M sulfuric acid. Following this, plates were read using a dual wavelength mode with a test wavelength of 450 nm and a reference wavelength of 620 nm. ELISA experimental data were plotted using the nonlinear least-squares analysis tool from Prism to fit a sigmoidal function ($\log_{10}$ IgG dilution and A450). From each sigmoidal curve, an $EC_{50}$ was extracted, corresponding to the IgG dilution that gives half of maximum absorbance.

## Categorizing the sustained IgG response

Categorization of patients into sustained IgG response categories (Low, Medium and High) is based on daily IgG titers, as dictated by the IgG $EC_{50}$ values during hospitalization. Patient data associated with short hospitalization stays (fewer than seven days) were eliminated from this analysis, to avoid categorizing the sustained IgG response of patients for which there were insufficient time points. IgG $EC_{50}$ values at any given day were averaged using a 5-day sliding window. Following this, all daily $EC_{50}$ values (from all patients while hospitalized) were pooled together, the 25[th] ($-\log_{10}(EC_{50}) = 2.91$) and 75[th] ($-\log_{10}(EC_{50}) = 4.47$) percentiles were used as the lower and upper thresholds respectively to categorize the sustained IgG response of each patient. Patients are required to have IgG $EC_{50}$ values for at least for 5 consecutive days within a particular range delimited by the IgG $EC_{50}$ thresholds in order to be assigned a particular category: i) low (sustained $EC_{50} \leq 2.91$); ii) medium (sustained $EC_{50} > 2.91$ and $\leq 4.47$) and; iii) high (sustained $EC_{50} > 4.47$). In order to minimize potential categorization errors due to varying IgG titers towards the end of the hospitalization stay (e.g. patients with sustained medium IgG titer showing higher IgG titers at the end of the hospitalization), the sustained IgG titer is required to extend to the end of the hospitalization stay. In addition, patients whose IgG titers fluctuated between the established thresholds, not reaching stable IgG titers within a delimited range for at least five days, were categorized as "non-conclusive".

## Longitudinal trajectory analysis

Longitudinal data on a clinical variable (IgG $EC_{50}$, WBC, neutrophils, lymphocytes, eosinophils, platelets, CRP and BUN) were averaged using a sliding window width of 5 days (e.g. IgG

$EC_{50}$ value for day 11 was averaged using all available IgG $EC_{50}$ readouts from days 9 to 13). Trajectories were analyzed with the scikit-learn and seaborn packages [67]. Line plots describe the median value and the 90% confidence intervals. Day-to-day statistical significance between groups (with $\geq$ 10 patients per group) was assessed using Mann-Whitney test, p-values were corrected for multiple hypothesis testing using the Benjamini-Hochberg correction [72].

## Classifiers

For each classification task, we used three types of models, all implemented using the scikit-learn package in Python (Supp File 3). We used random forests with num_trees set to 10, logistic regression with max_iter set to 10000, multilayer perceptron neural networks with 1 hidden layer of size 50, a regularization alpha of .1, and 1000 iterations for training. We also implemented a classifier which always predicts the most prevalent label in the training set to represent a baseline model which represents a 'common-sense' prior. We trained each model using 100 cross-validation iterations using the default settings from the 'train_test_split'function. We computed ROC and plot precision recall (PRC) curves using the 'plot_roc_curve'and 'plot_precision_recall_curve'functions and interpolating the resulting values. We used a linear interpolation method for ROC curves and a right-sided step function interpolation for the PRC curves (using the 'kind = 'next''parameter of the 'interp1d'function in sklearn). We generated the ROC and PRC plots by drawing confidence intervals around each point in the interpolated curves across the 100 iterations and shading in the regions spanning two standard errors of the mean. For feature extraction, we used the 'feature_importances_'and the 'coef_'attributes of the 'RandomForestClassifier'and 'LogisticRegressionClassifier'classes respectively.

## Admission values classifier

For the model which predicted outcomes using admission lab values and demographics, we used a combination of values from patient intake data, demographic information, and admissions lab values. Binary values that indicated the presence or absence of a symptom/condition included: fever, shortness of breath, sore throat, chills, diarrhea, oxygen supplementation required at presentation, African American (indicated in the EMR as 'Black' vs 'non-Black'), male sex, whether or not the patient resided in assisted living facilities, and whether or not the patient had glucose-6-phosphate dehydrogenase (G6PD) deficiency. Numeric features included age, BMI, initial temperature, initial respiratory rate, albumin levels, admissions BUN, admissions neutrophil count, admissions lymphocyte count, admissions aspartate aminotransferase (AST, a liver function test), alanine aminotransferase (ALT, a liver function test), admissions alkaline phosphatase level (ALK, a liver function test), LDH, (a biomarker of tissue damage), CRP,(a biomarker of inflammation), international normalized ratio (INR, a measure of blood clotting), admissions platelet count, admissions partial thromboplastin time (PTT, a measure of blood clotting), admissions prothrombin time (PT, a measure of blood clotting), admissions eosinophil count, and admission WBC count. All numeric columns were normalized to have mean zero and unit variance; binary values such as 'fever'were simply represented as a 1 or 0. Missing values were interpolated using the mean value of that column in the dataset.

## Longitudinal classifier

For the model which predicts outcomes in the future using present lab values, we broke down our longitudinal trajectories into a unique data point for each day. For interpolation, we rolled forward the last recorded value of a lab to represent the notion of 'latest lab value'. Our model

used 5 features: lymphocytes, WBC count, neutrophil count, IgG titer, and day of observation PSO. We normalized each feature by subtracting out the mean and dividing by the variance. We then sampled training/test splits on a per-patient basis; 25% of the patients were chosen for the test set and 75% to train each time. All samples associated with a patient were assigned to the respective data set. This strategy was adopted to prevent leaking of information between training and test set due to a single patient having similar labs and outcome status on different days (eg. a patient may be intubated for multiple days with the same lab values recorded throughout. A sample from one day may be allocated to the training set and another to the test set, making it possible for the model to memorize values at training time to unfairly boost performance). For evaluation of ROC/PRC on a daily/weekly basis, we used only those samples in the test set which were from a given day/week. All curves were plotted as an average across cross-validation runs.

## Shapley values

For each cross-validation cycle of the random forest longitudinal classifier, we computed Shapley values for the test set using the python package SHAP. Because we randomly resampled training/test sets during each training iteration, each sample only showed up in the test set for some fraction of the iterations. We kept track of this occurrence across the cross-validation and aggregated the Shapley values for each sample across the iterations it appeared in and averaged those values. This approach gave us a mean Shapley value for each sample. We then compared these values to the actual lab values for those samples to generate scatter plots and regressions. For clustering, we used the python package umap-learn to perform a UMAP clustering of our mean Shapley values. We then used the DBSCAN algorithm as implemented in the scikit-learn python package with eps set to .75 (the maximum distance between samples in a neighborhood) to cluster the samples. For each cluster, we performed a hypergeometric test (p-values were corrected for multiple hypothesis testing using Benjamini/Hochberg) to evaluate the enrichment of survivors and non-survivors.

## Statistics

Statistical parameters, including inclusion and exclusion criteria and statistical assumptions (normal distribution) are reported in the main text figures and figure legends. To calculate significance involving more than 2 groups, we have applied a Mann-Whitney test where p-values were corrected for multiple hypothesis testing using the Benjamini-Hochberg correction. Data are judged to be statistically significant when $p < 0.05$ in and are denoted with asterisks as followed: $^*p < 0.05$, $^{**}p < 0.01$. Statistical comparison on the length of hospitalization was assessed using a Kruskal-Wallis test. Association between categorical data (ie: distribution of sustained IgG titers or severity of disease) was evaluated by a Chi-square test of independence.

## Supporting information

**S1 Fig. The sustained IgG response of COVID-19 patients can be categorized based on the daily $EC_{50}$ titers.** $EC_{50}$ titers over time were used to categorize patients with $\geq 7$ days of hospitalization (n = 130) into three categories that describe the sustained IgG titer: (A) low, (B) medium and (C) high. Categorized patients are required to show daily IgG $EC_{50}$ titers for at least for 5 consecutive days within the range delimited by the $25^{th}$ and $75^{th}$ $EC_{50}$ percentiles: i) low IgG: $-\log_{10}(EC_{50}) \leq 25^{th}$ perc (grey background); ii) medium IgG: $25^{th}$ perc $> -\log_{10}(EC_{50}) \leq 75^{th}$ perc (blue background); iii) high IgG: $-\log_{10}(EC_{50}) > 75^{th}$ perc (blue background). We define "day post-symptom onset" (PSO) as the day relative to the patient-

reported onset of symptoms.
(TIFF)

**S2 Fig. Patients with sustained low IgG titers against SARS-CoV-2 S protein also show low IgG titers against SARS-CoV-2 N protein.** Comparison of IgG Ab response against SARS-CoV2 spike protein to nucleocapsid (N) protein. Anti-N and S IgG were measured at two serum dilutions and compared among hospitalized patients with a sustained low IgG response (patients = 17, serum samples = 23). ELISA experiments were performed in duplicates. Boxes extend from the 25$^{th}$ to 75$^{th}$ percentiles, the whiskers represent the minimum and maximum values and the middle line corresponds to the median.
(TIFF)

**S3 Fig. Anti-SARS-CoV-2 IgG titers upon hospital admission are not associated with COVID-19 severity or mortality.** (A) Admission IgG titers for survivors (cyan) and non-survivors (red). No significance difference was observed between groups (Mann-Whitney p = 0.4). (B) Fraction of survivors and non-survivors that tested positive for antibodies ($-log_{10}EC_{50} > 2.5$) the day of hospital admission (Purple: seropositive, Grey: seronegative). No significant association was observed (Chi-square p = 0.53). (C) Admission IgG titers for COVID-19 patients by clinical severity: mild (blue); moderate (green); severe (yellow) and; non-survivors (red). No statistical significance was observed (pairwise Dunn's multiple comparisons p>0.05). (D) Fraction of patients by severity that tested positive for antibodies ($-log_{10}EC_{50} > 2.5$) the day of hospital admission (Purple: seropositive, Grey: seronegative). No significant association was observed (Chi-square p = 0.51). (A, C) Boxes extend from the 25$^{th}$ to 75$^{th}$ percentiles, whiskers extend to the lowest and highest data point within 1.5 interquartile range of the lower and upper quartiles, the middle line corresponds to the median. IgG positivity threshold is indicated with a horizontal black dotted line at $-log_{10}(EC_{50}) = 2.5$.
(TIFF)

**S4 Fig. Deceased patients show lower anti-SARS-CoV-2 S IgG titers during an early time window (days 2–11 PSO).** Box-plot at days 0–14 PSO comparing IgG titers, as dictated by the corresponding $-log_{10}(EC_{50})$, for survival (cyan) and non-survival (red) patients. Boxes extend from the 25$^{th}$ to 75$^{th}$ percentiles, whiskers extend to the lowest and highest data point within 1.5 interquartile range of the lower and upper quartiles and the middle line corresponds to the median. The size of each group is described under each boxplot. Statistical significance is denoted with asterisks (Mann-Whitney; *p < 0.05, **p < 0.01). We define "day post-symptom onset" (PSO) as the day relative to the patient-reported onset of symptoms.
(TIFF)

**S5 Fig. Mildly ill patients consistently show lower IgG titers during a late time window (days 18–23 PSO).** Box-plot at days 15–26 PSO associating IgG titers, by the $-log_{10}(EC_{50})$, by outcomes: surviving patients requiring maximal oxygen supplementation (nasal canula: grey, non-rebreather mask: blue; high-flow: yellow; intubation: pink) and non-survival patient (red). Boxes extend from the 25$^{th}$ to 75$^{th}$ percentiles, whiskers extend to the lowest and highest data point within 1.5 interquartile range of the lower and upper quartiles. The size of each group is described under each boxplot. Statistical significance is denoted with asterisks (Mann-Whitney; *p < 0.05, **p < 0.01). We define "day post-symptom onset" (PSO) as the day relative to the patient-reported onset of symptoms.
(TIFF)

**S6 Fig. Non-surviving COVID-19 patients reach maximum IgG titers later than survivors.** Day (PSO) at which IgG plateau is reached for survivors (cyan) and non-survivors (red).

Plateau day for a particular patient is defined as the first day at which IgG titers reach 95% of the maximum IgG titer reported for that particular patient. Boxes extend from the 25th to 75th percentiles, whiskers extend to the lowest and highest data point within 1.5 interquartile range of the lower and upper quartiles, the middle line corresponds to the median. Statistical significance is denoted with asterisks (Mann-Whitney; $^*p < 0.05$).
(TIFF)

**S7 Fig. IgG trajectory during hospitalization based on COVID-19 patients' race.** (A) IgG $EC_{50}$ values (averaged value based on a five day sliding window) during the length of hospitalization of patients with different races: Black patients (black), White patients (pink). IgG positivity threshold is indicated with a horizontal black dotted line at $-\log_{10}(EC_{50}) = 2.5$. Shaded areas correspond to 90% confidence intervals. We define "day post-symptom onset" (PSO) as the day relative to the patient-reported onset of symptoms. (B) Mosaic plot of COVID-19 disease (mild, moderate, severe, non-survival) relative to race (Black, White). (C) Mosaic plot of IgG sustainable responses relative to race (Black, White). There was no statistical differences between race and severity, mortality or sustained IgG response.
(TIFF)

**S8 Fig. IgG trajectory during hospitalization based on COVID-19 patients ethnicity.** (A) IgG $EC_{50}$ values (averaged value based on a five day sliding window) during the length of hospitalization of patients with different ethnicity: Hispanic and non-Hispanic (pink). IgG positivity threshold is indicated with a horizontal black dotted line at $-\log_{10}(EC_{50}) = 2.5$. Shaded areas correspond to 90% confidence intervals. We define "day post-symptom onset" (PSO) as the day relative to the patient-reported onset of symptoms. (B) Mosaic plot of COVID-19 disease (mild, moderate, severe, non-survival) relative to Ethnicity (Hispanic, non-Hispanic). (C) Mosaic plot of IgG sustainable responses relative to Ethnicity (Hispanic, non-Hispanic).
(TIFF)

**S9 Fig. Features extracted from electronic medical records at the day of admission enable the prediction of severe disease and mortality.** Roc and Precision-Recall curves resulting from the evaluation of (A-B) random forest, (C-D) logistic regression and (E-F) neural network to predict COVID-19 severity and mortality based on Electronic Medical Records (including clinical, other laboratory data and demographics; see S7 Fig at the day of admission. Shaded areas correspond to ± 2 standard error of the mean. Legends describe the corresponding area under the curve.
(TIFF)

**S10 Fig. Importance of features extracted from electronic medical records upon day of admission to predict severity of disease and mortality.** Features include clinical, other laboratory data and demographics. Each feature is ranked according to its corresponding importance score obtained by random forest to predict a particular category of disease, including non-survival: (A) mild; (B) moderate; (C) severe; and (D) non-survival.
(TIFF)

**S11 Fig. Time-dependent clinical features enable prediction of mortality but not intubation.** Evaluation of multiple machine learning methods to predict mortality and intubation using longitudinally-monitored clinical data. (A-B) Predicting mortality five days into the future; (C-D) Predicting intubation five days into the future (purple: neural network, dark blue: Logistic regression, light blue: Random forest, red: Prevalence). (E-F) Predicting intubation five days into the future at different hospitalization weeks using a neural network (green: week two, brown: week three, black: week four). Shaded areas correspond to ± 2 standard

error of the mean. Legends described the corresponding area under the curve.
(TIFF)

**S12 Fig. Shapley values highlight the differential relationship between IgG titers and mortality in early and late PSO.** Plots describe the mean Shapley value Vs IgG titer at specific days PSO. We define "day post-symptom onset" (PSO) as the day relative to the patient-reported onset of symptoms.
(TIFF)

**S13 Fig. Hospitalization stay becomes more informative when predicting mortality.** The plot shows the mean Shapley value compared to the day PSO for each sample in the dataset. We define "day PSO" (PSO) as the day relative to the patient-reported onset of symptoms.
(TIFF)

**S14 Fig. Clustering of patient samples based on feature attributions reveal the importance of IgG titers, WBC, neutrophil and lymphocyte counts to predict survival among hospitalized COVID-19 patients.** Comparison of clusters corresponding to patient samples obtained at any given day during hospitalization (see Fig 6E). Each single day of any given patient is defined by the corresponding Shapley values on immunological features monitored longitudinally. Boxes extend from the 25th to 75th percentiles, whiskers extend to the lowest and highest data point within 1.5 interquartile range of the lower and upper quartiles.
(TIFF)

**S1 Table. Symptoms and complication cohort characteristics.** Data are shown as number and percentage, n (%). Mild category is constituted of those who did not require supplemental oxygen. Moderate category is constituted of patients who required nasal canula (1-4L/min to maintain SpO2 >92%) or non-rebreather mask. Severe category is constituted of patients who were on non-invasive ventilation, high-flow oxygen (≥6L/min to maintain SpO2 >92%), invasive mechanical ventilation or extracorporeal membrane oxygenation (ECMO). Non-survival patients are those who deceased during the course of hospitalization. RR, respiratory rate, BP, blood pressure, AMS, altered mental status, PO, per os (given by mouth), AKI: acute kidney injury; ARDS: acute respiratory distress syndrome.
(DOCX)

**S1 File. Trajectory statistics.**
(XLSX)

**S2 File. Trajectory statistics.**
(XLSX)

**S3 File. Machine learning.**
(PDF)

**S4 File. Shapley regression.**
(DOCX)

## Acknowledgments

The authors thank all the patients who consented to participate in our study and their health care providers during their hospitalization. We also thank Mimi Kim, Xiaonan Xue, Kenny Ye, Tao Wang and Li Xia for fruitful discussions on data analysis. We are grateful to Francesca La Carpia and Emily Happy Miller for useful comments during manuscript writing.

## Author Contributions

**Conceptualization:** Gorka Lasso, Saad Khan, Stephanie A. Allen, Libusha Kelly, Johanna P. Daily, Olivia Vergnolle.

**Data curation:** Gorka Lasso, Saad Khan, Stephanie A. Allen, Margarette Mariano, Jose A. Quiroz, Gregory Quevedo, Aditi Hegde, Avinash Malaviya, Ahmed Khokhar, Olivia Vergnolle.

**Formal analysis:** Gorka Lasso, Saad Khan, Stephanie A. Allen.

**Funding acquisition:** Kartik Chandran, Jonathan R. Lai, Libusha Kelly, Johanna P. Daily.

**Investigation:** Stephanie A. Allen, Margarette Mariano, Catalina Florez, Jose A. Quiroz, Gregory Quevedo, Ryan J. Malonis, George I. Georgiev, Karen Tong, Natalia G. Herrera, Nicholas C. Morano, Scott J. Garforth, Olivia Vergnolle.

**Methodology:** Margarette Mariano, Erika P. Orner, Ariel S. Wirchnianski, Robert H. Bortz, III, Ethan Laudermilch, M. Eugenia Dieterle, J. Maximilian Fels, Denise Haslwanter, Rohit K. Jangra, Olivia Vergnolle.

**Project administration:** Olivia Vergnolle.

**Resources:** Catalina Florez, Erika P. Orner, Aldo Massimi, Ryan J. Malonis, George I. Georgiev, Karen Tong, Natalia G. Herrera, Nicholas C. Morano, Scott J. Garforth, Ethan Laudermilch, M. Eugenia Dieterle, J. Maximilian Fels, Denise Haslwanter, Rohit K. Jangra, Jason Barnhill, Steven C. Almo, Kartik Chandran, Jonathan R. Lai.

**Supervision:** Jason Barnhill, Steven C. Almo, Kartik Chandran, Jonathan R. Lai, Libusha Kelly, Johanna P. Daily, Olivia Vergnolle.

**Validation:** Gorka Lasso, Saad Khan, Stephanie A. Allen, Margarette Mariano, Catalina Florez, Aldo Massimi, Aditi Hegde, Ariel S. Wirchnianski, Robert H. Bortz, III, Avinash Malaviya, Ahmed Khokhar, Olivia Vergnolle.

**Visualization:** Gorka Lasso, Saad Khan, Stephanie A. Allen, Olivia Vergnolle.

**Writing – original draft:** Gorka Lasso, Saad Khan, Olivia Vergnolle.

**Writing – review & editing:** Gorka Lasso, Saad Khan, Catalina Florez, Nicholas C. Morano, Denise Haslwanter, Jonathan R. Lai, Libusha Kelly, Johanna P. Daily, Olivia Vergnolle.

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
