## [Decision Letter · Decision Letter 0]

17 Sep 2021

Dear Dr vergnolle,

Thank you very much for submitting your manuscript "Longitudinally Monitored Immune Biomarkers Predict the Timing of COVID-19 Outcomes" for consideration at PLOS Computational Biology.

As with all papers reviewed by the journal, your manuscript was reviewed by members of the editorial board and by several independent reviewers. In light of the reviews (below this email), we would like to invite the resubmission of a significantly-revised version that takes into account the reviewers' comments.

We cannot make any decision about publication until we have seen the revised manuscript and your response to the reviewers' comments. Your revised manuscript is also likely to be sent to reviewers for further evaluation.

Sincerely,

Amber M Smith

Associate Editor

PLOS Computational Biology

Virginia Pitzer

Deputy Editor-in-Chief

PLOS Computational Biology

Reviewer's Responses to Questions

**Comments to the Authors:**

Reviewer #1: The manuscript on longitudinally immune biomarkers predict the timing of COVID-19 outcomes addresses an important subject of death by organ failure with a respiratory virus.

The study utilizes a well characterized data set on 149 individuals hospitalized in New York from March 1 to June 1.

The group has previously published on 103 patients given convalescent plasma from April to May 2020. https://insight.jci.org/articles/view/142270

Lacking from this paper is use of plasma, remdesivir and steroids which influences outcome in this population. In table 1 traditionally comorbid conditions are compared between groups rather than within a group. Notably immunosuppression is 30% in mild, 15% in moderate, 12% in severe and 23% in nonsurvivors. Please add this analysis. Also under residency 113 and 35 =148 rather than 149 in total and 42 and 15=57 rather than 58 in moderate. Another typo is severe length of stay as median 19 with IQR 23 and 25. Might be 13 to 25??

Another big point not explained is the spread amongst the groups of 130 analyzed for IGG which is the main point of the paper. Most likely more than half of the mild cases will be dropped. This reviewer would like to see the numbers in the groups for the 130 in mild, moderate and severe and nonsurvivors. This could be added to Table 1. Another important point is the number of antibody negative patients in each group and what impact being antibody negative on admission has directly on the mortality.

Another point that needs to be addressed is with exclusion of patients with less than 7 days of hospitalization most everyone is moderate to severe who is left for analysis. Mortality should be the single focus with so few mild cases in the analysis. For fig 2 F why only 77 cases and 109 in 2E. What was the reason for the dropouts.

The interesting finding was the low IGG titers in nonsurvivors at day 4 and then high at day 18 which is median day of death.

Another very confusing point is the switch back and forth of day form onset which is terrific to have but when used interchangeable with day of hospitalization. This reviewer prefers both descriptors to be used for point of reference.

Can the authors attribute the increase in WBC and neutrophils to multiorgan failure versus pure respiratory death in Fig 4. What percent of nonsurvivors had multiorgan failure on day 18?

The cluster analysis with Shapley values is an important aspect of the work. Please note how it performs on Day 4 and day 18 which were previous benchmarks. This was not clear from the discussion in result section and figure legends.

Supplement table 1. Why was pulse left out of vitals?

In fig 2 why were so few patients used in the first few days and more later? 22 to 85 ands 4 to 29 from day 0 to day 14

Supplement fig 7- can admission IgG, or IgM be added to list

Please point out in figure 10 and in other results that the biomarkers are significant as a group but not in a single patient. Meaning that a patient with high IgG and high WBC may very well live.

Reviewer #2: The authors use hospitalized COVID-19 patient data to predict disease outcome, in the form of oxygen support and mortality. Most importantly, longitudinal data during hospitalization (WBC, IgG titer, etc.) is used to make short-term predictions about mortality status, and the authors show that models trained on this data are more predictive that models that use only data from the time of admission. This is obviously an important study, and conceptually the approach makes sense to me. I do have one important suggestion to improve the analyses, and a bunch of small comments.

The "outcome" is also time-dependent. Currently the most severe outcome is used, but surely non-survivors will be in the "severe" category before they die, and "severe" patients might be categorized as "moderate" at an earlier time point. Can this additional information be used as well? This is most relevant in the longitudinal analysis, because the current disease status is not used as a predictor for the future status, while this is probably a very important predictor (see also my comment below). What kind of AUROC do you get if you include current disease status?

Comments:

Line 128: individuals with severe disease had a lower median age than moderate and non-survival. How can we explain the age difference between moderate and severe disease?

Line 143: one issue with the rolling five-day window is that you might include future data to make predictions. If you use observation X_t on day t, then X_t also contains information from days t+1 and t+2. This is relevant when you predict outcomes 1 and 2 days (and more) into the future.

Line 149: Why did you choose unequal group sizes for this classification? Why not use 33th and 66th percentile?

Line 155: 14.6% is the size of the Low group, right? So do you mean that a fraction of all patients had a Low response? That is because the groups are defined in this particular way.

Line 157: The way the cutoffs are defined is a bit puzzling. As I understand it, you take all IgG titers from all patients pooled together, and then define 25 and 75th percentiles as cutoff for Low, Medium, High. Then the highest sustained (>5d) value per patient is used to assign a group to a patient. This can be explained a bit better.

Line 162: That is, only if the IgG response is observed before the outcome is observed.

Line 171: One small remark: Because the maximum IgG per patient is used, those that stay in the hospital longer also have a higher chance of falling in the High category. So if the IgG titers were just random, you would find the same result. Of course they're not random, but perhaps you could test if the result still holds of you use the mean titer instead of the maximum.

Line 193: I don't quite understand the statement starting with "consequently". Why are IgG titers not associated with both severity AND mortality?

Line 196: refer to a figure. Is there a statistical analysis that confirms this?

Fig 4: WBC and Neutrophil for non survivers have a very specific shape (increase followed by decrease). Is there a relation with the times that patients died? Would it be possible indicate these times in the plot?

Line 237: A bit more words could be added to explain what the models do: is each model trained on one particular binary outcome? (moderate or non-moderate disease? mortality or survival?).

Line 265: Here you only mention mortality status, but later you also predict intubation status.

Figure 5B: the predictive performance increases with time since hospitalization (is that what you mean by length of hospitalization?). Is this because during the first week not many patients died, and hence there is nothing to predict?

Line 268: Have you also tried including the current disease status as a predictor? I can imagine that this could be significant, and it is also easily available.

Line 282: For which analysis are you doing this? The longitudinal or static data? The quadratic relation can be understood given that early, low IgG is associated with mortality, but also high late IgG. Is it possible to plot this together with the day of measurement to confirm this? Perhaps color the dots to indicate time?

Line 468: The information here requires that the reader has experience with these packages. Is there a way to explain the settings in a more general way? For instance, what does regularization alpha of .1 do? Also, I think that you should definitely publish your scripts (on github or a similar platform), because it should be easy for others to repeat these analysis with their own data.

Reviewer #3: Overall comments:

COVID-19 data suggests an intrinsic relationship between disease progression, severity of symptoms and immune markers. In this work, the authors measure longitudinally immune biomarkers of 147 hospitalised COVID-19 patients (e.g. neutrophils, lymphocytes, IFN). By examining their hospitalised patient measurements, they find survivors have a higher basal IgG titer at the day of hospital admission, compared to non-survivors. Patients who died during their hospitalisation had much lower IgG titers from days 2-11 compared to survivors. In addition, they find a clear delineation between survivors and non-survivors based on the longitundinal measurements of lymphocytes, neutrophils, and white blood cells. These results are not surprising as there are other studies of longitudinal data that echo results, however, it is still of interest to see these results confirmed in this study.

To investigate the relationship between biomarkers and fatality, the authors developed a machine learning framework trained on the longitudinally monitored immune biomarkers of hospitalised patients. They show that the framework can predict mortality more accurately than demographic and clinical data obtained upon hospital admission, i.e. using longitudinal measurements predicts mortality more accurately than models using single initial hospitalisation measurements. This also doesn’t feel surprising, as it suggests we can make more accurate predications with more data. While the work in this manuscript is very interesting, I feel the authors have a few areas that need addressing. I’ve detailed my comments/queries below.

Major

• In Fig 1A, it is not clear to me from the caption or the reference in the text why there are multiple data points for an individual patient corresponding to the yaxis label “Hospitalisation day after symptom onset”. Also, what are the clinical features that are colouring the heatmap? Is it disease severity or something else? The colour scheme also makes it very difficult to determine which patients experienced certain clinical features at certain times, i.e. to see the light red vs the dark red. I think this figure needs to be improved for clarity and needs more explanation. I would have thought that “hospitalisation day after symptom onset” should be a single measurement or data point I assume as patients would only come to hospital once?

• “Current models to predict COVID-19 severity rely on static features obtained upon hospitalised” – a quick literature search seems to bring up a lot of models that have used longitudinal hospiltalisation to parameterise their model and predict disease severity see for examples:

o https://doi.org/10.1093/cid/ciaa574

o DOI: 10.7554/eLife.64827

o https://doi.org/10.1186/s13054-020-03255-0

o https://doi.org/10.1371/journal.ppat.1009753

o Doi:10.1002/psp4.12574

And there were more. I think a more comprehensive literature search needs to be done as well as more background given to what is currently being done in this space as there seems to be other relevant works out there. In turn, it would be good for the authors to point out how their work differs to previous studies.

• “The trend of higher antibody responses in patients with severe symptoms compared to lower antibody responses in patients with mild-to-moderate symptoms” is an interesting finding. Is there any references or other clinical measurements that support this finding and does it have a biological explanation?

• Figure 2E and then the results in Fig 3 suggest that non-survivors experienced low sustained IgG symptoms (especially earlier), whereas severe disease patients experienced sustained IgG symptoms. Can this be explained? Or a biological explanation hypothesised?

• Leading on from the last comment, there is no reference to how the ML techniques utilised in this work have been used elsewhere and how reliable these are as a predictive tool. I think this should be addressed for readers so that they have a point of reference for the computational model presented.

• “Machine Learning models trained on longitudinally monitored immune features outperformed models trained on static features when predicting fatal outcome”. Fatal outcome is just one possible outcome, how can these models be better used to predict optimal therapeutic procotols?

• Is 147 a significant sample size for us to infer results? In turn, could this be a reason that it isn’t possible to distinguish markers for severe-mild patients in most cases?

• In Fig 6 where “for IgG, we observed a quadratic relationship where both high and low values were associated with a higher propensity of fatal outcomes relative to a middle value” is this statistically significant, as there seems to be a large variation in the Shapley value for the IgG EC50. In turn, what is the “middle value”? Can this be given more specifically?

• Is longitudinal data collection of hospitalised patients a feasible clinical protocol? And is there a way that the authors could determine if there are crucial cut-off/threshold cut-off times for which actual predications can be made for a particular patient. E.g. if a patient is exhibiting the markers of fatal disease after 5 days can we then say with some surety their disease progression and suggest intervention protocols? I think this may be what Figure 5C is showing but could more be expanded on this significance and how it could be used clinically.

Minor

• Is there a motivation for the choice of the different immunological measurements? E.g. why weren’t cytokine measurements taken?

• In Fig 1B, could you label the Color bar of the Choropleth map in the figure not just in the caption?

• I think a legend like that on in Supplementary Figure 3 would be helpful for Figure 3C

• In “…non-survival patients from days 18 to 23 (Fig 3C-D and Supp Fig 3)” it says “Fig 3C-D” but I can’t see a Fig 3D? Is this also the right reference as Fig 3C seems to be distinguishing the patients based on whether they required different oxygen supplies, not on their survival?

• A legend for Fig 4 that is on the figure would be helpful for the reader, so they don’t have to read the caption to determine what the colours mean

• In “… status within the next k days using immune biomarkers that would have been available on a given day, as we had these data available (Fig 5B-D).” Should this just be Fig 5C-D as Fig 5B seems to be about week by week ROC on a neural network. If not then it needs to be clearer in the text what Figure 5B is.

• I’m confused on how the prediction of “death and intubation” is represented in Figure 5, is this in a combined statistic? Or is just one of them represented in Fig 5? If so which one?

• Is there a label for the Fig 5C color bar?

• What do the red diamonds represent in Figure 6E?

• In Fig 4, the mild dynamics become strange between day 11 and 23, is this because the cohort has been reduced to only 1 patient? The reason for the data nonlinearity should be noted somewhere in the text.

Reviewer #4: The authors investigate the use of different biomarkers to model severe disease outcome for COVID-19, including mortality. This study is novel, as it uses longitudinal data of multiple biomarkers to predict mortality and intubation.

The longitudinal approach that the authors take is both interesting and promising. The manuscript is well written and clear for most parts.

I do have some major remarks with this work:

1. The dataset used is quite small, which is understandable, as it is harder to collect longitudinal data. However, unless I missed it, I believe there was no power analysis conducted in this work. I understand that this would also be non-trivial, especially for the longitudinal classifier, yet, some measure of uncertainty on the ROC curves could improve this (e.g. by computing the ROC for models from different test/train splits?).

2. Did the authors consider the effect of the imbalance in data between mortality and survival?

3. My main concern with this work is about the longitudinal classifier. I did not find that this model was well explained and I believe that a formal description is necessary. From the current description it was not clear how the trajectory for a patient up until time t, is passed to the model. As all patients have a different trajectory (i.e., they were admitted and left the hospital at different times), I wonder how this is taken into account. It is also not clear how the temporal relationship between the datapoints is modelled. The authors mention the use of a 2-layer neural network, but I wonder whether a LSTM or a temporal convnet was considered, to incorporate the temporal nature of the data.

4. The authors consider a temporal analysis of the longitudinal classifier, which I think is very useful. However, it would be interesting to include a measure of uncertainty in the predictions, and show how this uncertainty changes as more data becomes available, during a patient’s individual trajectory.

5. Adjacent to the remark about the temporal analysis, I was not fully certain whether all data collected up to the point that is to be predicted, is taken into account. Can the authors confirm this and perhaps clarify this in the manuscript? If this is not the case, I worry that the increasing prediction power with respect to mortality (Fig 5B) can be caused by the increasing likelihood of mortality, the longer individuals stay in the hospital.

6. The authors state “However, there is a need for orthogonal approaches that leverage longitudinal information to inform the early assessment of fatality risk in patients that do not yet show severe signs of disease (35).”, with which I agree. Yet, the models that were fitted on early data (1 week) exhibit poor performance (ROC curves Fig 5 B), so this does not really support this need? Furthermore, the patient cohort considered here exists out of individuals that were already hospitalized. Therefore, I assume that they show severe signs of disease already, so there are no experiments to backup up predicting disease severity in patients with no signs of severe disease?

7. There was one result I could not wrap my mind around: Fig 5 panel A vs B. In A the red curve corresponds to predicting mortality using static features, while in B the blue curve corresponds to predicting mortality using biomarkers from the first week. While I understand that these models are not exactly the same, yet, I would expect similar performance, as in panel B (1 week) you are using “more” data than in A (just admission data).

8. From a technical perspective, on line 470, you mention the number of iterations used. Did you also conduct a convergence analysis to support these values?

Minor remarks:

- One line 213, the authors report a set of biomarkers, please explain based on which criteria this selection was made.

- I found the fact that the methods section was at the end of the paper a bit inconvenient, as I needed to go back and forth while going to the results section. Perhaps some restructuring, or some repetition in the results section could help.

**Have the authors made all data and (if applicable) computational code underlying the findings in their manuscript fully available?**

Reviewer #1: Yes

Reviewer #2: **No: **Data and scripts are not provided.

Reviewer #3: Yes

Reviewer #4: **No: **The source code of the experiments is available on GitHub. There is also a data folder in the GitHub repo, but I did not find a statement whether all data can be found there.

PLOS authors have the option to publish the peer review history of their article (what does this mean?). If published, this will include your full peer review and any attached files.

Reviewer #1: No

Reviewer #2: No

Reviewer #3: No

Reviewer #4: No
---

## [Decision Letter · Decision Letter 1]

20 Dec 2021

Dear Dr vergnolle,

We are pleased to inform you that your manuscript 'Longitudinally Monitored Immune Biomarkers Predict the Timing of COVID-19 Outcomes' has been provisionally accepted for publication in PLOS Computational Biology.

Best regards,

Virginia E. Pitzer, Sc.D.

Deputy Editor-in-Chief

PLOS Computational Biology

Virginia Pitzer

Deputy Editor-in-Chief

PLOS Computational Biology

Reviewer's Responses to Questions

**Comments to the Authors:**

Reviewer #1: The authors are to be commended for a careful reply to comments.

The manuscript is much improved and adds tremendous value to the literature.

Reviewer #3: n/a

Reviewer #4: The authors have address my concerns.

**Have the authors made all data and (if applicable) computational code underlying the findings in their manuscript fully available?**

Reviewer #1: Yes

Reviewer #3: None

Reviewer #4: None

PLOS authors have the option to publish the peer review history of their article (what does this mean?). If published, this will include your full peer review and any attached files.

Reviewer #1: **Yes: **David J Sullivan

Reviewer #3: No

Reviewer #4: No

---

## [Editor Report · Acceptance letter]

10 Jan 2022

PCOMPBIOL-D-21-01445R1 

Longitudinally Monitored Immune Biomarkers Predict the Timing of COVID-19 Outcomes

Dear Dr Vergnolle,

I am pleased to inform you that your manuscript has been formally accepted for publication in PLOS Computational Biology. Your manuscript is now with our production department and you will be notified of the publication date in due course.

With kind regards,

Zsofia Freund
